# Presynaptic morphology and vesicular composition determine vesicle dynamics in mouse central synapses

Laurent Guillaud*, Dimitar Dimitrov, Tomoyuki Takahashi*

Cellular and Molecular Synaptic Function Unit, Okinawa Institute of Science and Technology Graduate University, Onna-son, Japan

**Abstract** Transport of synaptic vesicles (SVs) in nerve terminals is thought to play essential roles in maintenance of neurotransmission. To identify factors modulating SV movements, we performed real-time imaging analysis of fluorescently labeled SVs in giant calyceal and conventional hippocampal terminals. Compared with small hippocampal terminals, SV movements in giant calyceal terminals were faster, longer and kinetically more heterogeneous. Morphological maturation of giant calyceal terminals was associated with an overall reduction in SV mobility and displacement heterogeneity. At the molecular level, SVs over-expressing vesicular glutamate transporter 1 (VGLUT1) showed higher mobility than VGLUT2-expressing SVs. Pharmacological disruption of the presynaptic microtubule network preferentially reduced long directional movements of SVs between release sites. Functionally, synaptic stimulation appeared to recruit SVs to active zones without significantly altering their mobility. Hence, the morphological features of nerve terminals and the molecular signature of vesicles are key elements determining vesicular dynamics and movements in central synapses.

*For correspondence: laurent. guillaud@oist.jp (LG); ttakahas@ oist.jp (TT)

**Competing interests:** The authors declare that no competing interests exist.

## Introduction

At chemical synapses, neurotransmitters such as glutamate are contained in synaptic vesicles (SVs) and released by exocytosis. After exocytosis, SVs are retrieved by endocytosis, refilled with neurotransmitter, and transported to release sites to be reused (*Heuser and Reese, 1973*). Although cellular and molecular mechanisms of exocytosis and endocytosis have been extensively studied (*Sudhof, 2004*; *Jahn and Fasshauer, 2012*), movements of SVs between endocytosis and exocytosis are least understood. Previous studies have reported a wide range of SV mobility, several orders of magnitude different in their diffusion coefficient (D), between different types or even in the same type of presynaptic terminal (*Holt et al., 2004*; *Rea et al., 2004*; *Jordan et al., 2005*; *Westphal et al., 2008*; *Kamin et al., 2010*; *Lee et al., 2012*). Within nerve terminals, at both hippocampal and neuromuscular synapses, SVs reportedly display low mobility (*Lee et al., 2012*; *Lemke and Klingauf, 2005*; *Gaffield and Betz, 2007*), whereas SVs traveling across distant hippocampal presynaptic boutons show higher mobility (*Lee et al., 2012*; *Darcy et al., 2006*; *Fernandez-Alfonso and Ryan, 2008*; *Staras et al., 2010*). It is also controversial whether stimulation can affect SV mobility (*Gaffield et al., 2006*; *Peng et al., 2012*) or not (*Betz and Bewick, 1992*; *Westphal et al., 2008*; *Kamin et al., 2010*).

In spite of the wealth of molecular and cellular knowledge on neurotransmission, much less is known about the mechanisms regulating SV trafficking and supply in nerve terminals, and only few factors, modulating SV movements, have been identified thus far. Synapsin-1 has long been identified as a tether anchoring and releasing SVs in a phosphorylation-dependent manner (*Llinás et al., 1985*) and is thought to regulate SV mobility in various nerve terminals (*Jordan et al., 2005*;

**eLife digest** In the brains of mammals, communication between cells called neurons is vital for learning and memory. Pairs of neurons communicate at junctions called synapses. At a synapse, the first neuron releases chemical messengers into the gap between the cells, which then bind to and activate receptor proteins on the surface of the second neuron. The chemical messengers are released from bubble-like packages called synaptic vesicles that fuse with the first neuron's membrane and empty their contents into the synapse. This same neuron then retrieves and reassembles the components of the vesicle, ready to be filled again with the chemical messengers.

Neurons must continually retrieve and refill vesicles in order to continue transmitting information at synapses. But while the mechanisms of vesicle fusion and retrieval are well characterized, it remains unclear what triggers the movement and supply of vesicles inside synapses or how these processes are regulated. Deciphering these mechanisms will help us better understand how synapses work in healthy as well as diseased brains.

Using high-resolution microscopy, Guillaud et al. have now studied the movements of fluorescently labeled vesicles inside mouse brain synapses grown in the laboratory. This revealed that synaptic vesicles move in much more varied and complex ways than previously thought. The movement of vesicles changed depending on the type and developmental stage of the synapses. It also depended on the identity of particular proteins within the membranes of the vesicles themselves. These proteins, known as transporters, enable vesicles to take up the chemical messengers. Vesicles with different transporters showed different patterns of movement. Disrupting components of the internal skeleton of the neuron – specifically protein filaments called microtubules – also disrupted vesicle movement. By contrast, changes in the activity level of the synapse had no such effect.

The next step is to determine exactly how these factors regulate the movement of vesicles at synapses. Studies can then examine whether these processes are disrupted in neurological disorders, in which communication at synapses is often impaired.

*Gaffield et al., 2006*; *Rothman et al., 2016*). Actin network has also been implicated in the regulation of fast trafficking of SVs between synaptic sites in central synapses (*Darcy et al., 2006*). The roles of another cytosketal element, microtubules (MTs) and their associated molecular motors are well established for axonal transport (*Hirokawa et al., 2009*, *2010*), but their contribution in the regulation of SV dynamics within nerve terminals remains to be elucidated. Mechanical factors such as membrane tensions are proposed to promote accumulation of SVs in nerve terminals (*Siechen et al., 2009*) or to increase their mobility (*Ahmed et al., 2012*). More recently, experimentally constrained models suggested that physical determinants such as hydrodynamic interaction and vesicle collision influence SV movements in presynaptic terminals (*Rothman et al., 2016*). However, biological factors and mechanisms underlying the diversity of SV mobility in nerve terminals remain poorly understood.

In mammalian central synapses, imaging studies of SV mobility have been restricted to small hippocampal synapses. Here, by taking advantage of our newly developed giant glutamatergic synapses formed in primary culture between mouse auditory brainstem neurons (*Dimitrov et al., 2016*), we visualized SVs using real-time confocal microscopy, and analyzed their movements by automatically tracking large populations of fluorescently labelled vesicles within presynaptic terminals. We then performed spatio-temporal analyses of more than 35,000 vesicle trajectories to quantify their intrinsic dynamic properties (i.e. maximum speed and track length), their modalities of displacement (i.e. diffusive or active motions), and their overall mobility (i.e. diffusion coefficient). We compared these parameters between giant calyceal terminals and small hippocampal presynaptic boutons, between morphologically mature and immature calyceal terminals, and between calyceal terminals over-expressing two distinct vesicular proteins, VGLUT1 or VGLUT2. We also tested the effects of pharmacological disruption of the microtubule (MT) network, as well as of various stimulation protocols on SV movements within calyceal terminals. Altogether, our results revealed several factors influencing SV mobility and trafficking in the CNS: the type of synapse (giant calyceal or small

conventional terminals), the spatial localization of vesicles within a terminal (intra-swelling or inter-swelling), and the molecular composition of vesicles (vesicular glutamate transporter subtypes). Our results also suggest that MTs play essential roles in inter-synaptic movements of SVs and that synaptic stimulation does not induce any appreciable increase in SV mobility.

## Results

### Mobility of synaptic vesicles in giant calyceal terminals

To visualize SVs by fluorescence confocal microscopy, primary cultures over-expressing GFP in presynaptic neurons were incubated in the presence of Q655-labeled synaptotagmin-2 antibody (Q655-Syt2) from 1 hr to overnight to allow spontaneous uptake of the fluorescent marker into SVs. After image acquisition, automatic spot detection of individual SVs, on 2D confocal section, was performed using IMARIS software (see Materials and methods). Q655-Syt2 fluorescent spots distributed throughout the entire calyceal terminal over-expressing GFP (*Figure 1A*, *Video 1*). The average number of labeled vesicles per terminal significantly increased with length of exposure to Q655-Syt2, reaching ~1500 SVs per terminal after 16 hr (*Figure 1B*). The use of synaptotagmin-2 antibody significantly increased the efficiency of SV labeling, in comparison to quantum dots only (*Figure 1—figure supplement 1A,B*). The fluorescence intensity distribution in 150 nm confocal spots for Q655-Syt2-labeled SVs was similar to the one from 40 nm FITC-beads (*Figure 1—figure supplement 1C*), suggesting single vesicle detection in our tracking experiments. Compared with Q655, the broader distribution of C5E fluorescence intensity results from the pH sensitivity of the dye emission.

Two different tracking algorithms were used to analyze their mobility within the terminal: a Brownian motion algorithm to detect non-directional diffusive movements and an autoregressive motion algorithm to detect both diffusive and active directional movements (*Video 2*). Analysis of vesicle trajectories, according to their maximum speeds and trajectory lengths, showed that Brownian motion algorithm often failed to trace trajectories with long directional runs, while autoregressive algorithm could trace both directionless random fluctuation and long directional runs (*Figure 1C and D*) with parameters determined empirically. The diffusion coefficient (D), calculated from the mean square displacement (MSD) curves of trajectories identified with the autoregressive algorithm ($D = 0.065 \pm 0.004$ μm$^2$/s), was seven times greater than that calculated from the MSD determined by the Brownian algorithm ($D = 0.009 \pm 0.001$ μm$^2$/s). This suggests that vesicle mobility could be underestimated by analysis based solely on the Brownian motion algorithm (*Figure 1E*). Thus, we used autoregressive algorithm for the tracking of SV trajectories in this study.

We next determined background mobility in our experimental conditions. Aldehyde fixation of labeled giant terminals lowered SV mobility ($D = 0.008 \pm 0.001$ μm$^2$/s) to the level of mobility observed for 40 nm fluorescent polystyrene beads adsorbed onto glass coverslips ($D = 0.008 \pm 0.001$ μm$^2$/s), indicating that the signal-to-noise ratio in our tracking experiments was ~8 (*Figure 1E*).

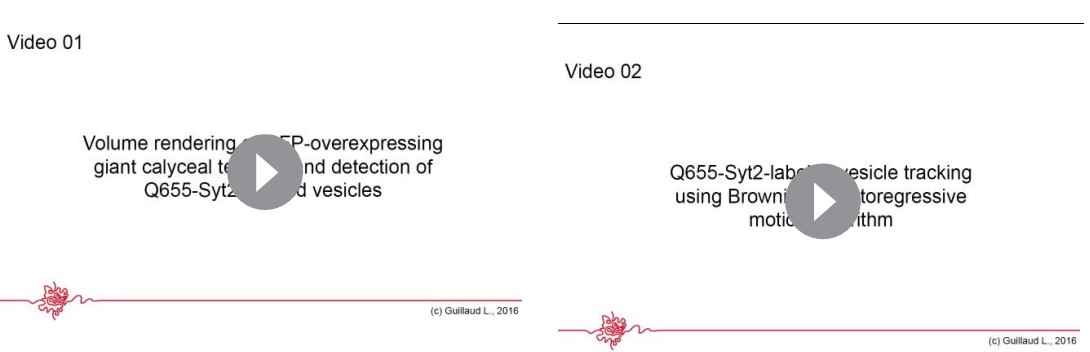

Video 01

Volume rendering of GFP-overexpressing giant calyceal terminal and detection of Q655-Syt2-labeled vesicles

(c) Guillaud L., 2016

**Video 1.** Volume rendering of GFP-expressing giant calyceal terminal and spot detection of Q655-Syt2-labeled SVs (color-coded according to their z position). DOI: 10.7554/eLife.24845.011

Video 02

Q655-Syt2-labeled vesicle tracking using Brownian or autoregressive motion algorithm

(c) Guillaud L., 2016

**Video 2.** Tracking of Q655-Syt2-labeled SVs using autoregressive (Red) or Brownian (Blue) motion algorithm. DOI: 10.7554/eLife.24845.012

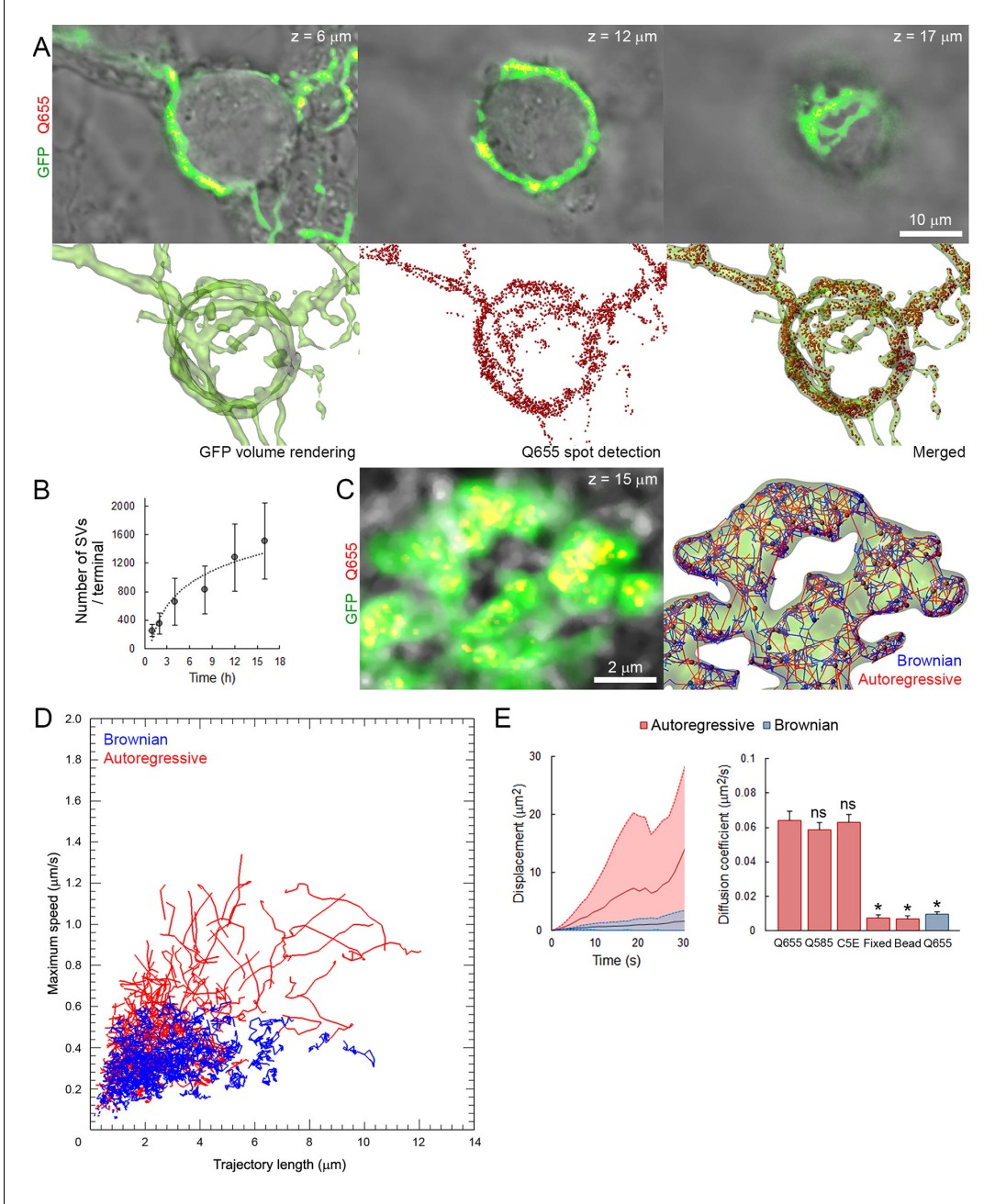

**Figure 1.** Autoregressive motion analysis reveals high and broad range of SV mobilites in cultured giant terminals. (**A**) Confocal z-stack imaging of a giant presynaptic terminal expressing cytosolic GFP- (Green) and Q655-Syt2 (Red)-labeled vesicles; corresponding volume rendering of GFP terminal and SV detection (see *Video 1*). (**B**) The number of labeled SVs detected in whole presynaptic terminal. (**C**) Live confocal imaging and SV tracking with the autoregressive motion (Red) or Brownian motion (Blue) algorithm (see *Video 2*). (**D**) Scatter plot of SV trajectory lengths and maximum speeds superimposed with individual trajectory traces, color-coded as in (**C**). (**E**) Comparison of SV displacements and diffusion coefficients for Q655-Syt2- (n = 12 terminals), Q585-Syt2- (n = 12) or C5E-Syt2- (n = 12) labeled vesicles, Q655-Syt2-labeled vesicles after chemical fixation (n = 12) and 40 nm beads (n = 12 ROI). Two-tailed unpaired t-test (*p<0.05; ns, not significant).

The following source data and figure supplements are available for figure 1:

**Source data 1.** Data and statistics for *Figure 1E*.

**Source data 2.** Data and statistics for *Figure 1—figure supplement 1B*.

*Figure 1 continued on next page*

*Figure 1 continued*

**Source data 3.** Data and statistics for *Figure 1—figure supplement 2D*.

**Source data 4.** Data and statistics for *Figure 1—figure supplement 3D*.

**Figure supplement 1.** Q655-Syt2 labels SV more efficiently than Q655 alone.

**Figure supplement 2.** Labeling and tracking of SVs with C5E-Syt2.

**Figure supplement 3.** Newly retrieved SVs have lower mobilities and restricted distributions in giant terminals.

To ensure that SVs loaded with Syt-2 antibody can undergo exocytosis, Syt-2 was labeled with the pH-sensitive fluorochrome CypHer5E (C5E) instead of quantum dots (Q655). C5E shows high-fluorescence intensity at low pH = 5.5 and low intensity at neutral pH, allowing us to monitor SV exo/endocytic cycles and to track endocytosed SVs. Labeling of SVs with C5E-Syt2 (*Figure 1—figure supplement 2A*) was identical to labeling of SVs with Q655-Syt2 (*Figure 1A*). Stimulation with 65 mM KCl induced an 80% decrease in the fluorescence intensity of C5E-Syt2-loaded vesicles, showing that a large proportion of labeled SVs underwent exocytosis (*Figure 1—figure supplement 2B*). Furthermore, diffusion coefficients of SVs labeled with Qdots (Q655 and Q585) or with C5E were similar (*Figure 1E*), indicating that labeling methods did not affect their dynamic or functional properties.

Using C5E-Syt2-labeled vesicles, we performed fluorescence recovery after photo-bleaching (FRAP) experiments (*Figure 1—figure supplement 2D,E*) to compare SV mobility with those measured from MSD (*Figure 1E*). Data pooled from three different terminals showed that in presynaptic swellings, ~40% of SVs were mobile ($T_{1/2}$ = 13.2 ± 1.1 s) with a calculated diffusion coefficient (*Axelrod et al., 1976*) of 0.029 ± 0.007 $\mu m^2$/s, whereas in finger-like processes, ~80% of SVs were mobile ($T_{1/2}$ = 6.3 ± 0.7 s) with a diffusion coefficient of 0.061 ± 0.007 $\mu m^2$/s. Although we observed lower D in swellings calculated by FRAP, D in fingers remained comparable to the value estimated by autoregressive MSD curves (*Figure 1E*).

In addition, it has been reported that the mobility of newly retrieved SVs in hippocampal synapses is highest after endocytosis and gradually decreases thereafter (*Kamin et al., 2010*). We examined whether it might also apply to giant synapses. Vesicles were first loaded with Q655-Syt2 by overnight incubation to achieve homogenous labeling of the resting vesicular pool. The next day, the same terminals were exposed to Q585-Syt2 for 1 to 3 hr and Q655- and Q585-Syt2-loaded vesicles were tracked (*Figure 1—figure supplement 3A*). One hour after the second loading, SVs newly labeled with Q585-Syt2 remained in restricted regions of the terminal, not mixing with previously retrieved Q655-Syt2-labeled vesicles (Pearson co-localization index, p=0.087 ± 0.06; *Figure 1—figure supplement 3B*), and showed relatively low mobility (D = 0.045 ± 0.001 $\mu m^2$/s). Two to three hours after the second loading, SVs labeled with Q585-Syt2 increased their mobility (D = 0.066 ± 0.005 $\mu m^2$/s after 3 hr; *Figure 1—figure supplement 3C,D*) and gradually mixed with the pre-existing pool of SVs labeled with Q655-Syt2 (p=0.266 ± 0.035; *Figure 1—figure supplement 3A,E*).

## Fast and heterogeneous vesicles mobility in giant calyceal terminals

We first characterized the basic properties of movements (maximum speed, trajectory length, modality of displacement, and diffusion coefficient) of SVs loaded with Q655-Syt2 within giant calyceal terminals. We categorized vesicle trajectories into three groups according to their individual trajectory lengths relative to the average size (~2.1 $\mu$m) of large presynaptic swellings (*Figure 2—figure supplement 1*): short (S) trajectories (length <2 $\mu$m, intra-swelling trafficking), medium (M) trajectories (2 $\mu$m < length < 4 $\mu$m, intermediate trafficking) and long (L) trajectories (4 $\mu$m < length, inter-swelling trafficking; *Figure 2A* and *Video 3*). During SV trafficking, longer trajectories were consistently accompanied by faster maximal speeds (*Figure 2B* and *Video 4*), and movements of SVs were highly heterogeneous between (*Figure 2B*) and within (*Figure 2—figure supplement 2A, B*) individual trajectories. Of all SVs examined in 12 giant terminals, 61% had short trajectories and slow motility, whereas 39% had long (up to 6 $\mu$m) and fast (up to 0.8 $\mu$m/s) directional movements

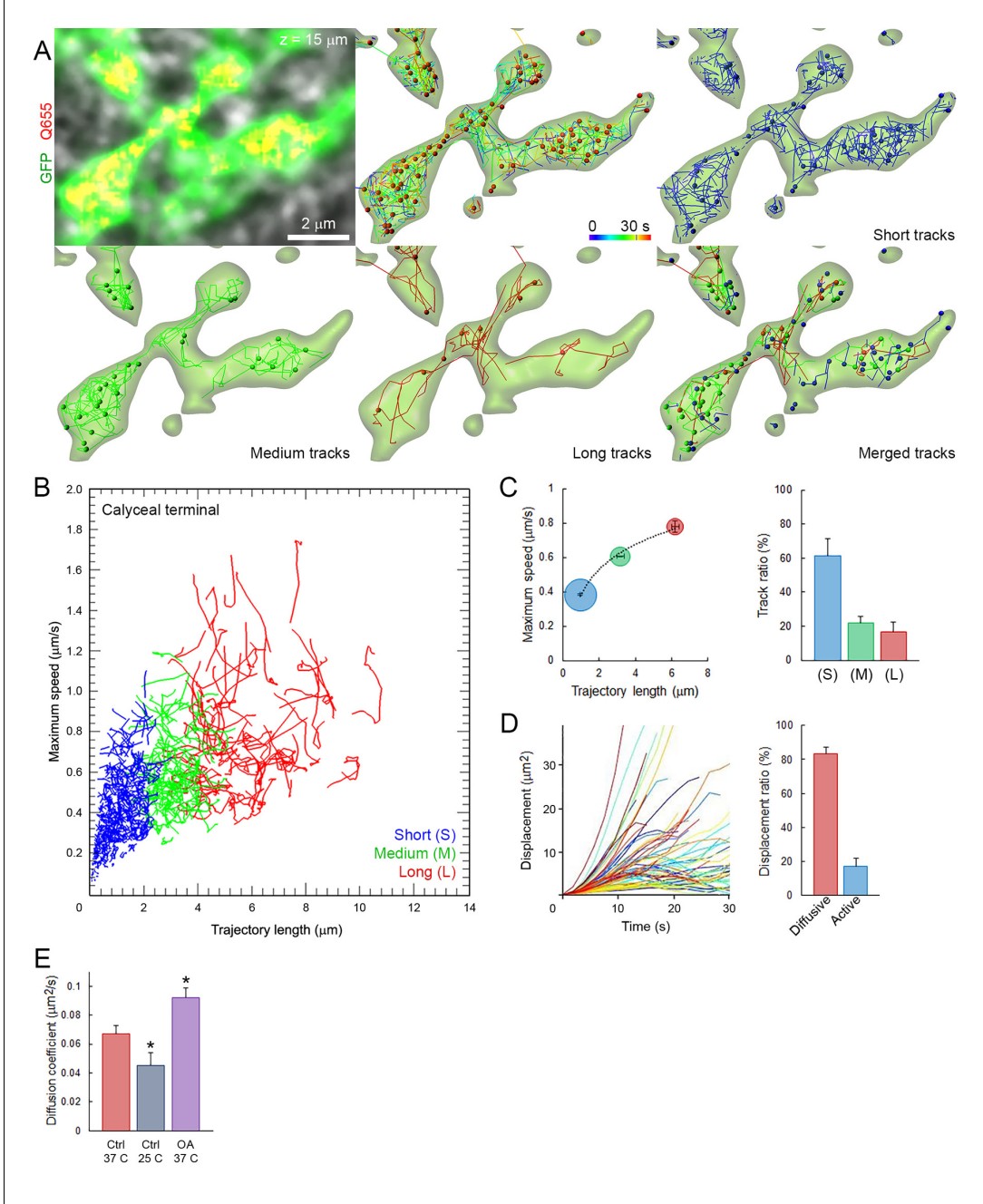

**Figure 2.** Fast and heterogeneous SV movements occur at giant calyceal synapses. (**A**) Live confocal imaging of a giant calyceal terminal expressing cytosolic GFP- and Q655-Syt2-labeled vesicles, with SV tracking, color-coded over time, or sorted according to trajectory lengths (Blue <2 μm, Green 2–4 μm and Red >4 μm, see **Video 3**). (**B**) Scatter plot of SV trajectory lengths and maximum speeds superimposed with individual trajectory traces (see **Video 4**), color-coded as in (**A**). (**C**) Classification and quantification of SV movements in three groups based on their maximum speed and trajectory length (n = 6175 trajectories). (**D**) Displacement curves and displacement modalities (Red: diffusive motion, Blue: active motion) of identified traces (n = 6175 trajectories). (**E**) Diffusion coefficient of SVs at 37° C (n = 12 terminals) or 25°C (n = 4); or in the presence of 2.5 μM OA at 37°C (n = 9). Two-tailed unpaired t-test (*p<0.05).

The following source data and figure supplements are available for figure 2:

**Source data 1.** Data and statistics for *Figure 2E*.
**Source data 2.** Data and statistics for *Figure 2—figure supplement 2E and H*.

*Figure 2 continued*

**Source data 3.** Data and statistics for *Figure 2—figure supplement 3B*.
**Figure supplement 1.** Comparison of calyceal and hippocampal cultures.
**Figure supplement 2.** Mobility and displacement modality of SVs in giant calyceal terminals.
**Figure supplement 3.** Data acquisition rate does not affect SV tracking.

(*Figure 2C*). We next analyzed displacement modalities of individual vesicles along their trajectories. Most SVs moving over long trajectories accelerate and decelerate, while SV speed remained more constant for medium and short trajectories (*Figure 2—figure supplement 2A,B*) resulting in a broad variety of displacements (*Figure 2D*). We categorized displacement modalities of SVs into two groups: one with diffusive motion and the other with active (facilitated or impeded displacements) motion (*Figure 2—figure supplement 2C*), and found that ~20% of labeled SVs move actively in calyceal terminals (*Figure 2D*).

We next examined effects of various factors known to influence SV mobility. At mouse hippocampal synapses (*Kraszewski et al., 1995*; *Jordan et al., 2005*) and frog neuromuscular junctions (*Betz and Henkel, 1994*; *Gaffield et al., 2006*), the phosphatase inhibitor, okadaic acid (OA) dramatically increases SV mobility. At giant calyceal terminals, OA increased SV mobility by ~43% (from $D = 0.065 \pm 0.004$ $\mu m^2/s$ to $0.093 \pm 0.008$ $\mu m^2/s$, *Figure 2E*), a significant, but moderate increase compared with those reported previously. As suggested from FRAP experiments (*Figure 1—figure supplement 3D,E*), SV mobility was ~1.4 times higher in finger-like processes ($D = 0.059 \pm 0.004$ $\mu m^2/s$) than in swellings ($D = 0.044 \pm 0.003$ $\mu m^2/s$, *Figure 2—figure supplement 2D,E*). Consistently, OA also enhanced SV mobility ~1.4 times in both regions (*Figure 2—figure supplement 2E*). Lowering the temperature from physiological (37°C) to non-physiological (25°C) conditions decreased the diffusion coefficient of SVs by ~30% (from $D = 0.065 \pm 0.004$ $\mu m^2/s$ to $D = 0.046 \pm 0.003$ $\mu m^2/s$, *Figure 2E*), with a temperature coefficient ($Q_{10}$) of 1.6, indicating that SV movements were moderately more temperature-dependent than diffusion ($Q_{10} = 1.3$).

We further analyzed SV movements and distributions in 3D within presynaptic terminals and found that under resting conditions, 25% of labeled vesicles move outward toward the postsynaptic cell, 14% move inward, and 35% move laterally near the synaptic cleft (*Figure 2—figure supplement 2F,G*). Trajectory proportion and diffusion coefficient were similar between 2D or 3D tracking (*Figure 2—figure supplement 2H*), suggesting a minimum bias in our tracking method cause by movements of different SV in and out of the focal plan. Four-fold changes in our data acquisition rate also did not significantly affect SV tracking and trajectory proportion (*Figure 2—figure supplement 3*), indicating that long SV trajectories are not likely resulting from the detection of different SVs sequentially moving throughout the focal plan.

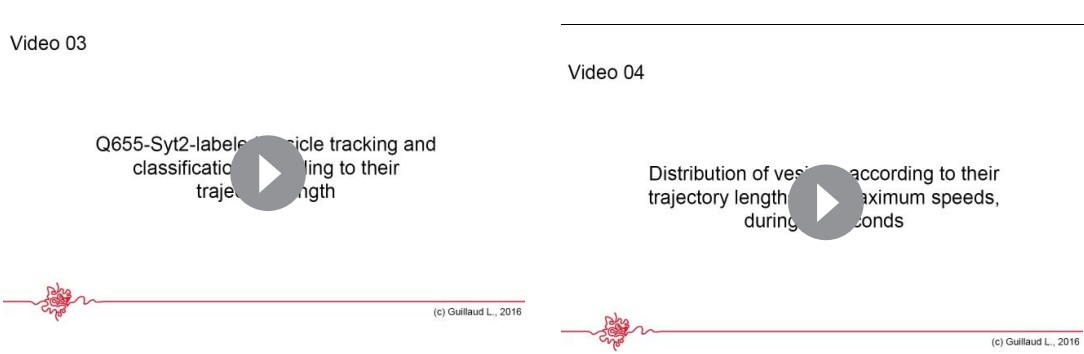

**Video 3.** Tracking of Q655-Syt2-labeled SVs color-coded according to SV trajectory lengths (Blue: short, Green: intermediate and Red: long trajectories).

**Video 4.** Scatter plot of SV trajectory lengths and maximum speeds during 30 s time-series.

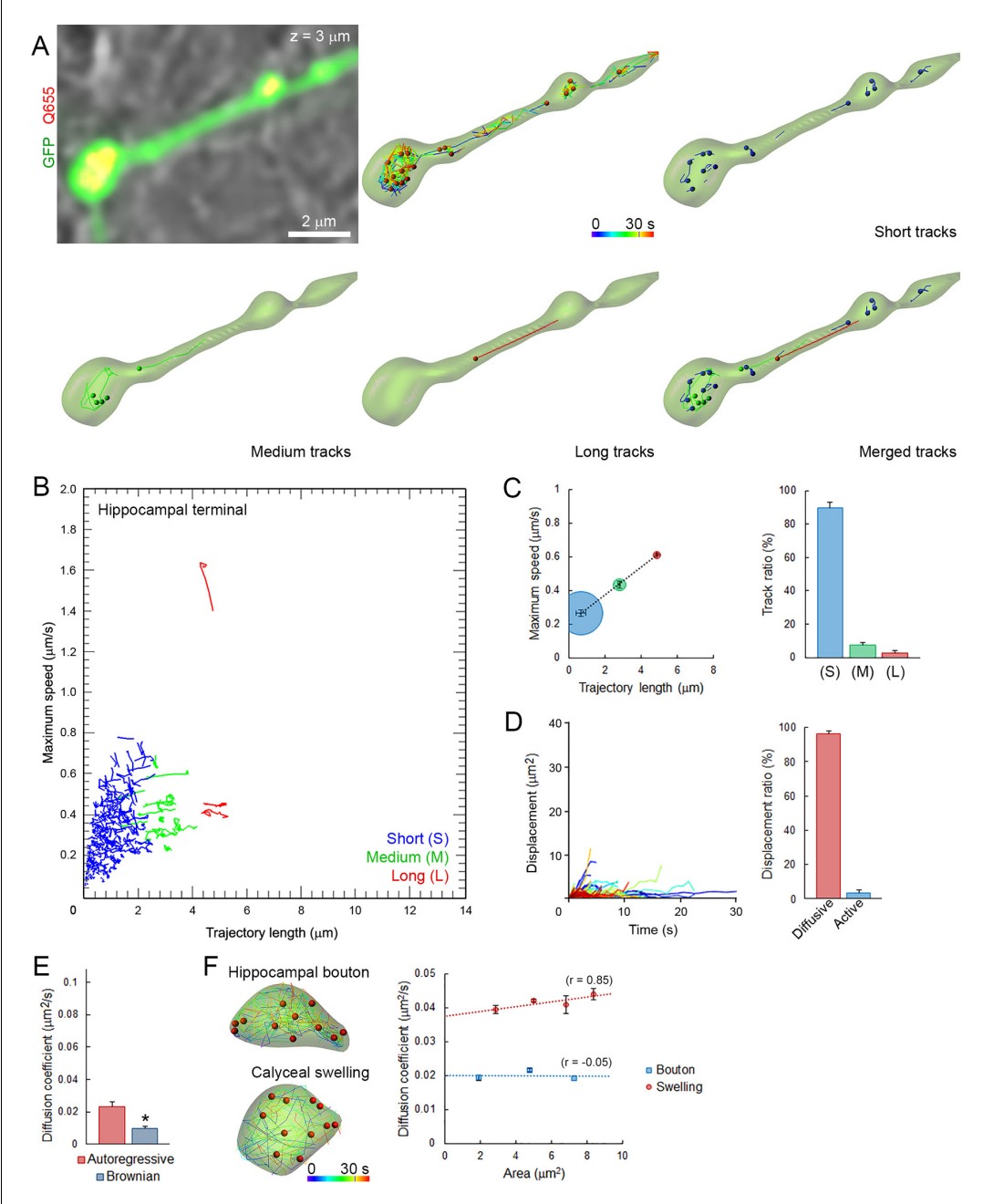

**Figure 3.** Small and homogeneous SV movements occur at small conventional synapses. (**A**) Live confocal imaging of a hippocampal bouton expressing cytosolic GFP- and Q655-Syt2-labeled vesicles, with SV tracking color-coded over time, or sorted according to trajectory lengths (Blue <2 μm, Green 2–4 μm and Red >4 μm). (**B**) Scatter plot of SV trajectory lengths and maximum speeds superimposed with individual trajectory traces, color-coded as in (**A**). (**C**) Classification and quantification of SV movements in three groups based on their maximum speed and trajectory length (n = 2958 trajectories). (**D**) Displacement curves and displacement modalities (Red: diffusive motion, Blue: active motion) of identified traces (n = 2958). (**E**) Diffusion coefficients of SVs in hippocampal terminals calculated from autoregressive (Red) or Brownian (Blue) analysis (n = 9 terminals). Two-tailed unpaired t-test (*p<0.05). (**F**) Comparison of SV mobility in hippocampal boutons (n = 9) or calyceal swellings (n = 9).

The following source data is available for figure 3:

**Source data 1.** Data and statistics for *Figure 3E*.

Altogether, these data demonstrate that movements of SVs in calyceal terminals were highly dynamic and heterogeneous under resting conditions.

## Dynamic properties and mobility of SVs in conventional-sized hippocampal terminals

We next applied the same imaging techniques and analytical methods used in giant calyceal terminals to small hippocampal terminals (*Figure 2—figure supplement 1A*). As observed in giant calyceal terminals, Q655-Syt2-labeled vesicles were distributed throughout hippocampal terminals overexpressing cytosolic GFP (*Figure 3A*). The distribution of trajectories according to their maximum speed and length from nine hippocampal terminals showed that SVs in small hippocampal boutons had predominately (89%) short trajectories and slow speeds (*Figure 3B and C*). We next compared modalities of displacements of SVs in small hippocampal boutons (*Figure 3D*) with those in giant calyceal terminals (*Figure 2D*). In hippocampal boutons, 97% of SVs moved by diffusion while only 3% of SVs moved actively. The diffusion coefficient of SVs in resting hippocampal boutons was $0.024 \pm 0.003$ $\mu m^2/s$ (*Figure 3E*), ~3 times lower than that in giant calyceal terminals ($D = 0.065 \pm 0.004$ $\mu m^2/s$, *Figure 2E*). No differences in D values, calculated using Brownian motion analysis, were observed between hippocampal ($D = 0.009 \pm 0.001$ $\mu m^2/s$, *Figure 3E*) and calyceal ($D = 0.009 \pm 0.001$ $\mu m^2/s$, *Figure 2E*) terminals.

To examine whether different SV mobility resulted from size differences of presynaptic terminals, we compared SV diffusion coefficients between calyceal swellings and hippocampal boutons. Although the size of calyceal swellings was about twice that of hippocampal boutons (*Figure 2—figure supplement 1B*), when SV movements were compared between swellings and boutons having similar areas, mobility of SVs in swellings was consistently ~2-fold higher than that in boutons (*Figure 3F*). Within calyceal terminals, SV mobility increased with the size of the swellings (r = 0.85), whereas there was no positive correlation (r = −0.05) between the size of boutons and SV mobility in hippocampal terminals. These results indicated that between two types of central synapses, with different terminal sizes and morphologies, SV mobility differs substantially; giant synapses show faster and more heterogeneous vesicle movements than small synapses. Thus, the type and the morphology of central synapses appear to significantly impact SV mobility in nerve terminals.

## Morphological development influences SV mobility in giant calyceal terminals

As at the developing calyx of Held, giant calyceal terminals in culture undergo significant morphological rearrangements and functional maturation, which were initially classified into four stages (*Dimitrov et al., 2016*). During morphological maturation of synapses, the volume and surface area of presynaptic terminals (*Figure 4—figure supplement 1A,B*), as well as the number of labeled vesicles (*Figure 4—figure supplement 1C*) increased gradually. Hence, we categorized developing giant terminals into two groups: 'immature' (stages 1 and 2) and 'mature' (stages 3 and 4), and compared SV dynamics between them. Immature terminals were characterized by prominent finger-like processes and presynaptic volumes below 1000 $\mu m^3$, whereas mature terminals were composed of numerous swellings interconnected with finger-like branches and presynaptic volumes above 1000 $\mu m^3$. Q655-syt2-labeled vesicles distributed throughout immature and mature terminals and their movements were analyzed in finger-like processes and interconnected swellings (*Figure 4A and B*). SV movements displayed wider heterogeneity in immature (trajectory length, 0–14 $\mu m$; maximum speed, 0–1.8 $\mu m/s$) than in mature (trajectory length, 0–9 $\mu m$; maximum speed, 0–1.4 $\mu m/s$) terminals (*Figure 4C*). The distribution of trajectories according to their maximum speed and length from nine immature terminals and nine mature terminals showed that morphological maturation of terminals significantly reduced the length (from $6.83 \pm 0.46$ $\mu m$ to $5.57 \pm 0.07$ $\mu m$) and speed (from $0.82 \pm 0.02$ $\mu m/s$ to $0.74 \pm 0.02$ $\mu m/s$) of SVs with long (L) trajectories (*Figure 4D*). The proportion of slow and short vesicles movements increased from immature (50%) to mature (70%) terminals, whereas fast, long directional movements decreased from 50% to 30% during morphological maturation of terminals (*Figure 4D*). As the calyceal terminal morphology advanced, diffusive motions relative to active directional motions increased by ~1.8 fold (*Figure 4E*). Consistently, SV mobility was significantly reduced after morphological development of calyceal terminals, from immature stage 2 ($D = 0.072 \pm 0.005$ $\mu m^2/s$) to mature stage 4 ($D = 0.052 \pm 0.003$ $\mu m^2/s$) terminals (*Figure 4F* and

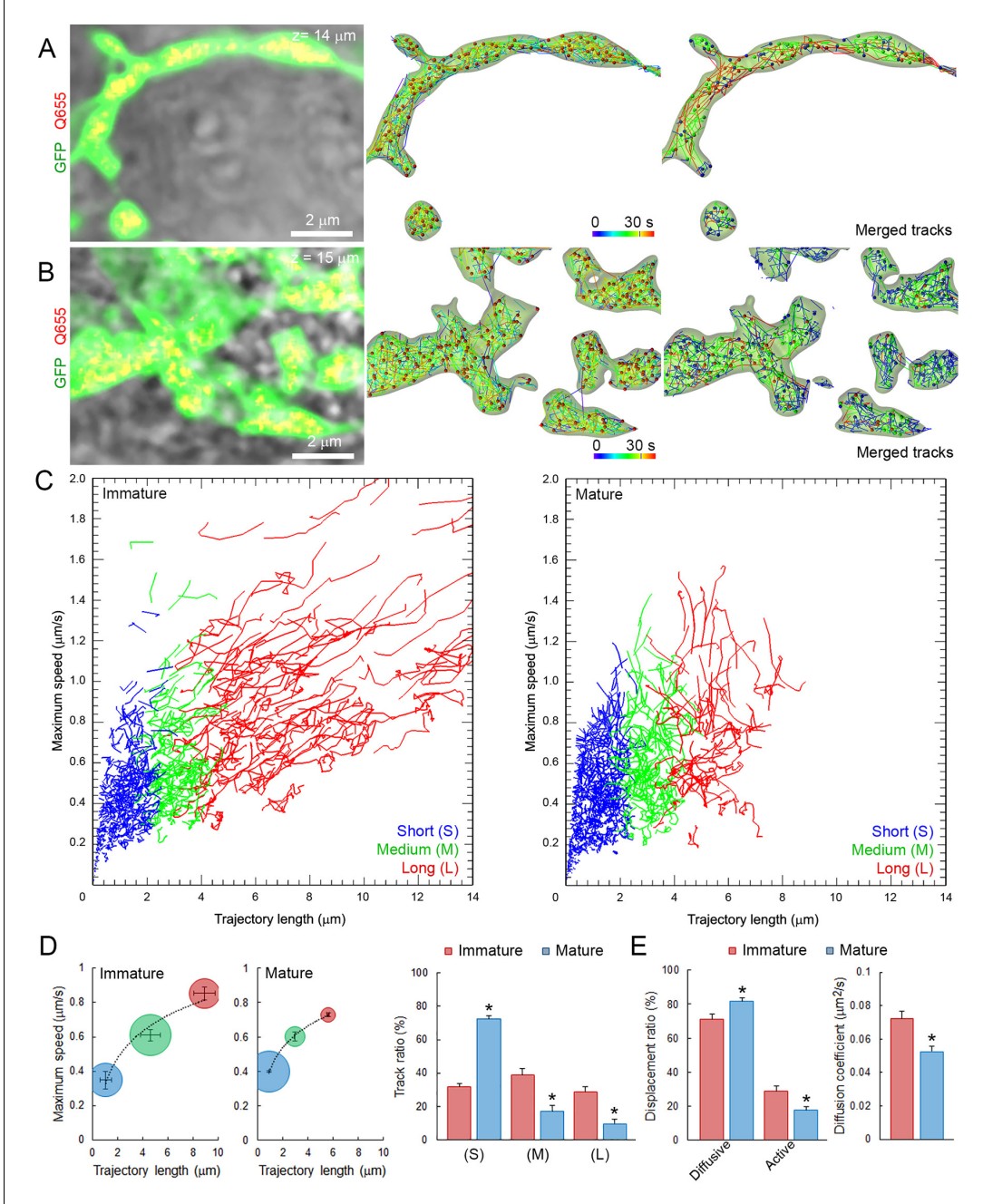

**Figure 4.** SV mobility decreases after morphological maturation of giant terminals. (**A**) Live confocal imaging of a giant immature terminal expressing cytosolic GFP and Q655-Syt2-labeled vesicles, with SV tracking color-coded over time, or sorted according to trajectory length (Blue <2 μm, Green 2–4 μm and red >4 μm). (**B**) Confocal imaging of a giant mature terminal as described in (**A**). (**C**) Scatter plot of SV trajectory lengths and maximum speeds superimposed with individual trajectory traces from immature (left panel) or mature (right panel) calyceal terminals, color-coded as in (**A**). (**D**) Classification and quantification of SV movements in three groups based on their maximum speeds and trajectory lengths in immature (Red, n = 9 terminals) and mature (Blue, n = 9) terminals. (**E**) Displacement modalities and diffusion coefficients of SVs in immature (Red, n = 9) and mature (Blue, n = 9) terminals. Two-tailed unpaired t-test (*p<0.05).

The following source data and figure supplement are available for figure 4:

**Source data 1.** Data and statistics for *Figure 4D and E*.

**Source data 2.** Data and statistics for *Figure 4—figure supplement 1D*.

*Figure 4 continued on next page*

*Figure 4 continued*

**Figure supplement 1.** Morphological maturation of giant terminals involves a developmental switch in SV mobility.

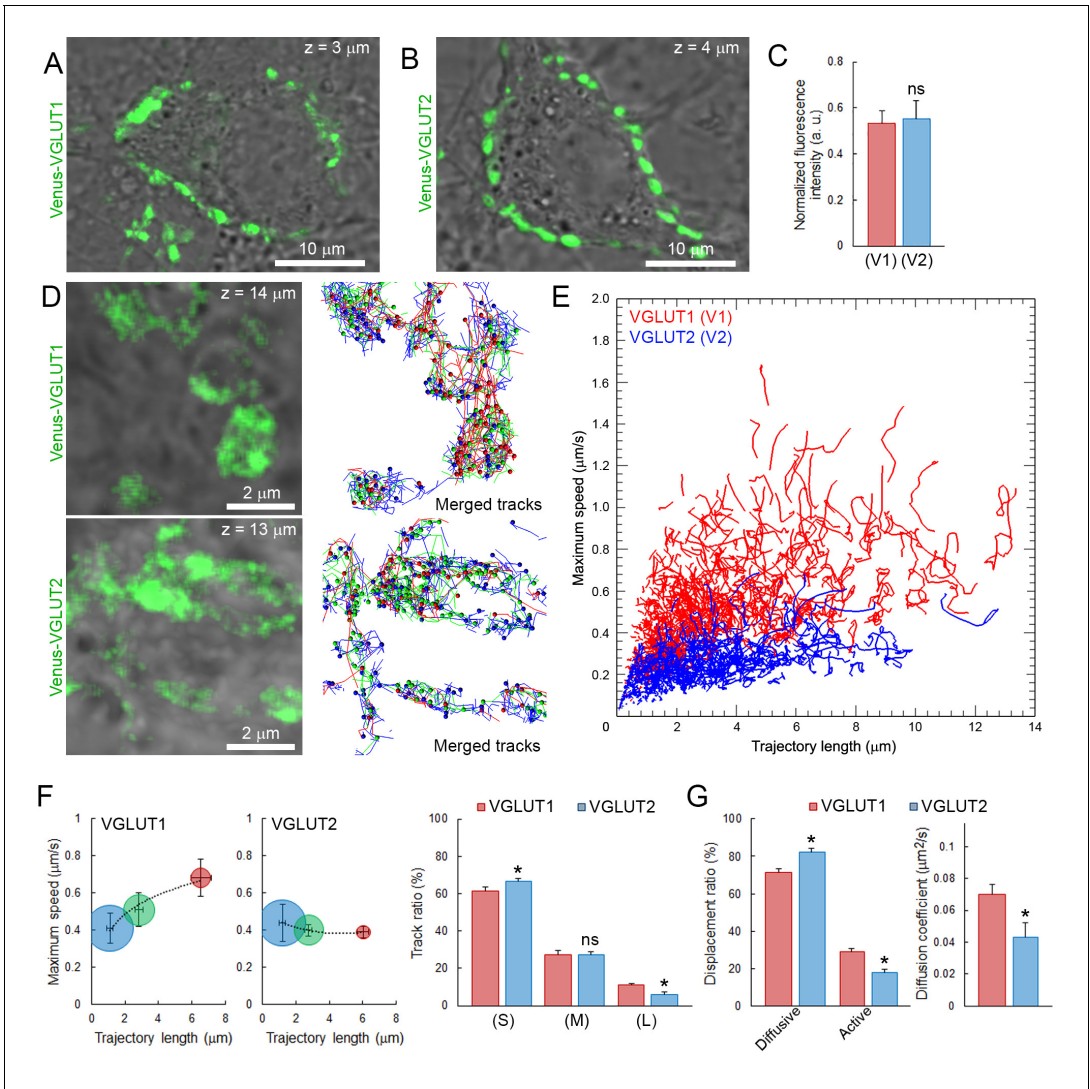

**Figure 5.** Vesicular glutamate transporter isoforms regulate SV dynamic properties. (A) Confocal z-stack imaging of giant presynaptic terminal expressing Venus-VGLUT1 (Green). (B) Confocal z-stack imaging of giant presynaptic terminal expressing Venus-VGLUT2 (Green). (C) Quantification of the fluorescence intensity of Venus-VGLUT1 (Red, n = 6 terminals) and Venus-VGLUT2 (Blue, n = 6) in giant calyceal terminals after 21 days in culture. (D) Upper panels: Live confocal imaging of a calyceal terminal expressing Venus-VGLUT1, and SV tracking sorted according to trajectory lengths (Blue <2 µm, Green 2–4 µm and red >4 µm). Lower panels: Live confocal imaging of a calyceal terminal expressing Venus-VGLUT2, and SV tracking sorted according to trajectory lengths (Blue <2 µm, Green 2–4 µm and red >4 µm). (E) Scatter plot of SV trajectory lengths and maximum speeds superimposed with individual trajectory traces in Venus-VGLUT1 (Red) or Venus-VGLUT2 (Blue) overexpressing terminals. (F) Dynamic properties of SVs expressing Venus-VGLUT1 or Venus-VGLUT2. (G) Displacement modalities and diffusion coefficients of Venus-VGLUT1 expressing vesicles (Red, n = 6) and Venus-VGLUT2 expressing vesicles (Blue, n = 6). Two-tailed unpaired t-test (*p<0.05; ns, not significant).

The following source data and figure supplement are available for figure 5:

**Source data 1.** Data and statistics for *Figure 5C, F and G*.

**Figure supplement 1.** Localization of endogenous vesicular glutamate transporter isoforms in cultured giant calyceal terminals.

 SV mobility transiently increases from stage 1 (D = 0.062 ± 0.002 µm$^2$/s) to stage 2 (D = 0.075 ± 0.004 µm$^2$/s) and stage 3 (D = 0.071 ± 0.002 µm$^2$/s), to finally decrease in stage 4 (D = 0.055 ± 0.001 µm$^2$/s). These results indicate that after a transient increase, SV mobility is globally down-regulated after the morphological maturation of giant calyceal terminals.

## Vesicular glutamate transporter subtypes influence dynamic properties and mobility of SVs

Vesicular glutamate transporters (VGLUTs) transport glutamate into synaptic vesicles (*Takamori et al., 2000*; *Fremeau et al., 2004*) and their subtypes VGLUT1 and VGLUT2 reportedly produce different release probabilities (*Weston et al., 2011*). As giant calyceal terminals in culture express both VGLUT1 and VGLUT2 (*Dimitrov et al., 2016*), we questioned whether expression of different VGLUT subtypes might affect dynamic properties of SVs in nerve terminals. Venus-VGLUT1 and Venus-VGLUT2 were overexpressed in different presynaptic neurons and both localized on SVs throughout giant calyceal terminals (*Figure 5A and B*), similar to the distribution of endogenous VGLUT1 and VGLUT2 (*Figure 5—figure supplement 1*). After 21 days in culture, expression levels of Venus-VGLUT1 and Venus-VGLUT2, deduced from their fluorescence intensity in the terminals, were similar (*Figure 5C*). Venus-VGLUT1- or Venus-VGLUT2-expressing SVs were visualized and their movements tracked (*Figure 5D*) and analyzed, as described for vesicles labeled with Q655-Syt2. The distribution of trajectories, according to their maximum speed and length, showed higher heterogeneity in movements of SVs expressing Venus-VGLUT1 than of SVs expressing Venus-VGLUT2 (*Figure 5E*). Analysis of trajectories, displacement modalities, and diffusion coefficients of vesicles pooled from six terminals over-expressing Venus-VGLTU1 and 6 terminals over-expressing Venus-VGLUT2 remarkably revealed that VGLUT1-containing SVs had wider ranges and faster movements than VGLUT2-containing SVs (*Figure 5E*). Surprisingly, the generally observed positive correlation between trajectory length and maximum speed was absent in VGLUT2-containing SVs, but was retained in VGLUT1-containing SVs (*Figure 5F*). VGLUT1-containing SVs displayed ~1.6 times (28.9%) more active displacements than VGLUT2-containing SVs (18.2%, *Figure 5G*). Furthermore, the mobility of vesicles expressing Venus-VGLUT1 (D = 0.072 ± 0.005 µm$^2$/s) was ~1.7 times higher than that of those expressing Venus-VGLUT2 (D = 0.043 ± 0.008 µm$^2$/s, *Figure 5H*). Assuming a one-to-one expression ratio in calyceal terminals, the average diffusion coefficient of Venus-VGLUT1 and Venus-VGLUT2 expressing SVs (D = 0.058 ± 0.006 µm$^2$/s) was similar to those of Q655-Syt2-labeled SVs (D = 0.065 ± 0.004 µm$^2$/s) or C5E-Syt2-labeled SVs (D = 0.063 ± 0.003 µm$^2$/s). These results indicated that expression of VGLUT1 conferred higher mobility upon SVs than expression of VGLUT2, and suggest that the molecular composition of vesicles can influence their dynamic properties.

## Involvement of microtubules in inter-synaptic trafficking

Roles of microtubules (MTs) are well established in axons (*Conde and Cáceres, 2009*), but poorly understood in nerve terminals. Our data indicate that a significant number of vesicles move within giant terminals with speed comparable to that of molecular motors along MTs involved in intracellular organelle transport (*Hirokawa et al., 2009*, *2010*). Synaptic growth involves reorganization of the neuronal cytoskeleton (*Roos et al., 2000*), which in turn could affect SV mobility. Hence, we examined whether cytoskeletal elements such as MTs and kinesins could be involved in transport of SVs in giant calyceal terminals. Immunofluorescence analysis revealed that giant presynaptic terminals were enriched in de-tyrosinated α-tubulin-containing polymers that interconnect neighboring and distant presynaptic swellings filled with SVs (*Figure 6A* and *Figure 6—figure supplement 1A*). The molecular motor, KIF1A (*Okada et al., 1995*), was also found in calyceal terminals and partially co-localized with the vesicular protein, synaptophysin, within swellings (*Figure 6—figure supplement 1B,C*). KIF1A also co-localized with VGLUT1-containing SVs in regions that interconnect swellings (*Figure 6—figure supplement 1D,E*), suggesting that KIF1A might carry SVs along MTs within and between swellings. To test the possible involvement of MT-based transport in SV mobility, we disrupted presynaptic MT networks by bath application of 30 µM nocodazole. Fluorescence intensity of Silicon-Rhodamine (SiR)-tubulin-containing polymers decreased by ~30% after 90 min incubation with nocodazole (*Figure 6B and C*), indicating that a significant fraction of MTs in calyceal terminals

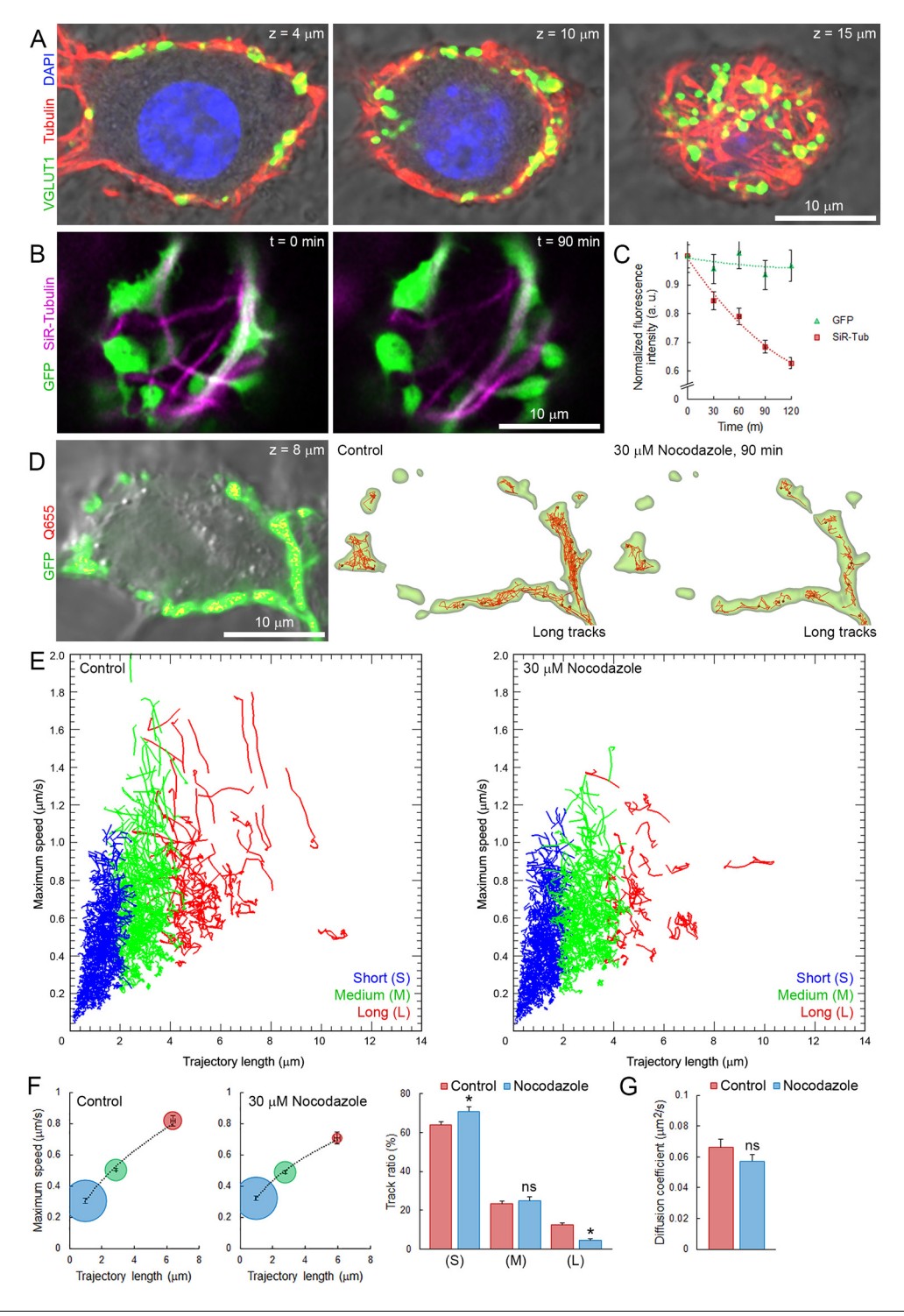

**Figure 6.** Presynaptic MT network regulates long and rapid directional SV movements. (**A**) Confocal z-stack imaging of a calyceal terminal labelled with antibodies against de-tyrosinated α-tubulin (Red), VGLUT1 (Green) and DAPI (Blue). (**B**) Live confocal imaging of a calyceal terminal over-expressing GFP and labeled with SiR-Tubulin before and after treatment with 30 µM Nocodazole. (**C**) Quantification of GFP- and SiR-Tubulin fluorescence intensity during nocodazole treatment. (**D**) Live confocal imaging of a calyceal terminal expressing cytosolic GFP and Q655-Syt2 labeled vesicles, and SV tracking (long tracks displayed only). (**E**) Scatter plot of SV trajectory lengths and maximum speeds superimposed with individual trajectory traces in control (left panel) or nocodazole-treated terminals (right panel), color-coded (Blue <2 µm, Green 2–4 µm and red >4 µm). (**F**) Classification and quantification of SV movements in three groups based on their maximum speeds and trajectory lengths in control (Red, n = 8 terminals) and nocodazole-treated (Blue,

*Figure 6 continued on next page*

Figure 6 continued

n = 8) terminals. (G) Diffusion coefficient of SVs in control (Red, n = 8) and nocodazole-treated (Blue, n = 8) terminals. Two-tailed unpaired t-test (*p<0.05; ns, not significant).

The following source data and figure supplements are available for figure 6:

**Source data 1.** Data and statistics for *Figure 6F and G*.

**Figure supplement 1.** Microtubules and kinesins localize in giant calyceal terminals.

**Figure supplement 2.** Actin network localizes in presynaptic swellings.

were depolymerized. Depolymerization of one third of MT networks was sufficient to significantly impair fast, long directional SV movements within terminals (*Figure 6D and E*). The distribution of trajectories according to their maximum speed and length (from eight untreated and eight nocodazole-treated terminals) showed that MT disruption reduced the proportion of SVs with long directional trajectories and concurrently increased the proportion of SVs with short displacements (*Figure 6F*). However, the apparently lower mobility observed after nocodazole treatment (D = 0.057 ± 0.004 μm$^2$/s) was not significant compared to controls before treatment (D = 0.066 ± 0.005 μm$^2$/s, *Figure 6G*). On the other hand, disruption of actin network, which essentially localized within discreet regions of presynaptic swellings (*Figure 6—figure supplement 2A,B*), did not affect the movement of SVs between swellings (*Figure 6—figure supplement 2C*). These results indicate that MT depolymerization selectively reduces the maximum speed and the proportion of SVs travelling long distances, suggesting a dominant role of MT-based transport for SVs undergoing inter-swelling trafficking in giant terminals.

## Effects of various stimulations on SV movements in giant calyceal terminals

It is controversial whether presynaptic stimulation increases SV mobility (*Peng et al., 2012*) or not (*Lemke and Klingauf, 2005*; *Kamin et al., 2010*; *Joensuu et al., 2016*). We examined this issue on C5E-Syt2-labeled SVs, using three different stimulation protocols; (i) bath-application of 65 mM KCl to induce massive exocytosis, (ii) bath-application of 500 mM sucrose to induce SV exocytosis from the readily releasable pool (RRP) (*Stevens and Tsujimoto, 1995*), and (iii) sustained electrical field stimulation at 1 Hz to trigger physiological SV exocytosis. In giant terminal swellings, KCl stimulation decreased the number of labeled SVs (*Figure 1—figure supplement 2B*) and redistributed SVs to smaller areas (*Figure 1—figure supplement 2C*). The overall distribution of SV trajectories did not change significantly after KCl stimulation (*Figure 7A*). However, long trajectories markedly decreased while the proportion of intermediate trajectories increased and short trajectories remained unchanged (*Figure 7D*). SV displacement modalities were also unaffected by KCl stimulation (*Figure 7E*). Surprisingly, SV mobility was reduced ~2.7 times (from D = 0.059 ± 0.004 μm$^2$/s to 0.022 ± 0.002 μm$^2$/s) after KCl stimulation (*Figure 7F*). However, this reduction was only observed in swelling regions, but not in finger-like regions (*Figure 2—figure supplement 2E*).

Bath application of hypertonic sucrose solution had no effect on SV trajectories (*Figure 7B*), nor on the proportion of long, short, and intermediate movements (*Figure 7D*). Interestingly, hypertonic sucrose induced a ~2.2-fold decrease in the proportion of active motions (from 14.2% to 6.3%, *Figure 7E*). However, hypertonic sucrose had no effect on SV mobility in the RRP (*Figure 7F* and *Figure 2—figure supplement 2E*). We next stimulated calyceal terminals electrically at 1 Hz. Although this stimulation induced significant endocytosis (*Figure 7—figure supplement 1A*), SV trajectories (*Figure 7B*), displacement modalities (*Figure 7C*), and overall mobility (*Figure 7F* and *Figure 2—figure supplement 2E*) remained unchanged.

It has been postulated that SVs undergoing spontaneous and stimulation-evoked release belong to different pools in presynaptic terminals (*Chung et al., 2010*). Thus, we investigated whether SVs, labeled spontaneously or during electrical stimulation, might show different dynamic properties. Electrical stimulation in the presence of C5E-Syt2, significantly increased the number of C5E-loaded vesicles in terminals, as well as their fluorescence intensity (*Figure 7—figure supplement 1A*), as

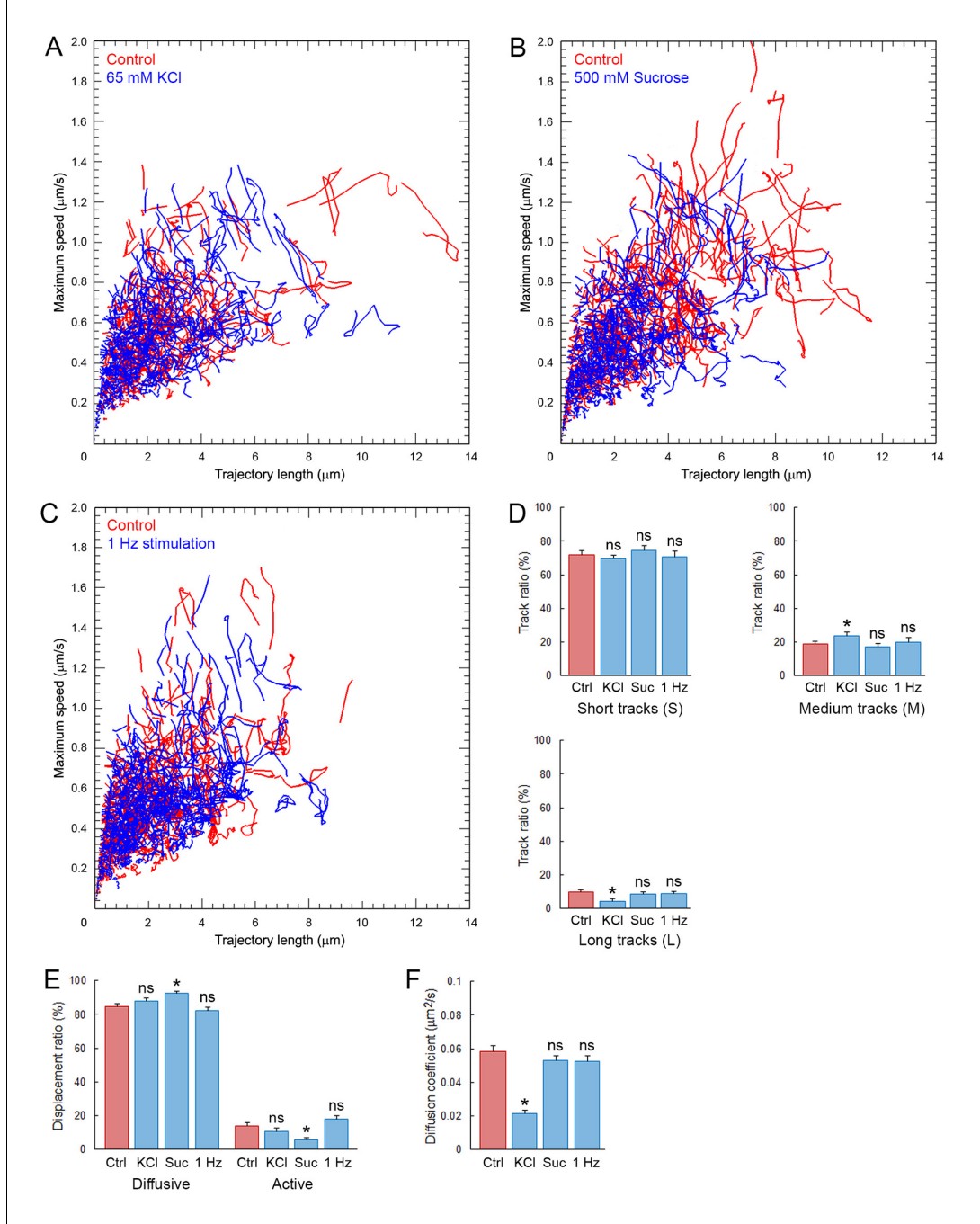

**Figure 7.** Synaptic stimulation does not increase SV mobility. Analysis of C5E-Syt2-labeled SVs in giant calyceal terminals. (**A**) KCl stimulation: Scatter plot of SV trajectory lengths and maximum speeds superimposed with individual trajectory traces in control terminals (Red) or terminals incubated with 65 mM KCl (Blue). (**B**) Sucrose stimulation: Scatter plot of SV trajectory lengths and maximum speeds superimposed with individual trajectory traces in control terminals (Red) or terminals incubated with 500 mM sucrose (Blue). (**C**) Electrical simulation: Scatter plot of SV trajectory lengths and maximum speeds superimposed with individual trajectory traces in control terminals (Red) or terminals during 1 Hz electrical field stimulation for 30 s (Blue). (**D**) Trajectory length analysis in control (Red) and KCl-treated terminals, sucrose-treated terminals, or 1 Hz-stimulated (Blue) terminals. (**E**) Displacement modality analysis in control (Red) and KCl-treated terminals, sucrose-treated terminals, or 1 Hz-stimulated (Blue) terminals. (**F**) Diffusion coefficient analysis in control (Red) and KCl-treated terminals, sucrose-treated terminals, or 1 Hz-stimulated (Blue) terminals. (KCl treatment: n = 6; sucrose treatment: n = 6; 1 Hz stimulation: n = 6 in D, (**E and F**). Two-tailed unpaired t-test (*$p<0.05$; ns, not significant).

The following source data and figure supplement are available for figure 7:

**Source data 1.** Data and statistics for *Figure 7D, E and F*.

*Figure 7 continued on next page*

*Figure 7 continued*

**Source data 2.** Data and statistics for *Figure 7—figure supplement 1C*.

**Figure supplement 1.** SV mobility does not change after spontaneous or stimulated uptake.

reported previously in KCl-stimulated hippocampal neurons (*Kraszewski et al., 1995*). However, the mobility of SVs loaded upon electrical stimulation (D = 0.052 ± 0.003 $\mu m^2$/s) was not different from that of those loaded spontaneously for 1 hr before stimulation (D = 0.048 ± 0.005 $\mu m^2$/s, *Figure 7—figure supplement 1B,C*). The mobility of SVs loaded with Syt2-C5E spontaneously or during synaptic stimulation was also consistent with the reduced mobility of newly endocytosed Q585-loaded SVs observed after 1 hr (*Figure 1—figure supplement 3D*), suggesting that the mobility of SVs labeled spontaneously or during stimulation were similar. Thus, while the number of moving vesicles might vary, neither chemical nor electrical stimulation significantly increased their dynamic properties and mobility in giant calyceal terminals.

Finally, we tested whether stimulation might affect SV dynamics near active zones (AZs). We labeled surface GluR1/2 on post-synaptic neurons to localize the release sites on the pre-synaptic terminal previously loaded with syt2-C5E (*Figure 8A*), and compared SV movements within and outside from putative AZs (*Figure 8B*). We showed that during field electrical stimulation at 1 Hz, the proportion of SVs significantly increased by ~1.8 times near release sites (from 21% to 38%, *Figure 8C*). However, the mobility of SVs inside ($D_{in}$ = 0.027 ± 0.008 $\mu m^2$/s) and outside ($D_{out}$ = 0.044 ± 0.003 $\mu m^2$/s) of AZs remained largely unaffected during stimulation ($D_{in}$ = 0.024 ± 0.007 $\mu m^2$/s and $D_{out}$ = 0.037 ± 0.005 $\mu m^2$/s; *Figure 8D*). The apparent lower mobility of SVs within AZs compared to SVs outside AZs was also not statistically significant before (p=0.08,

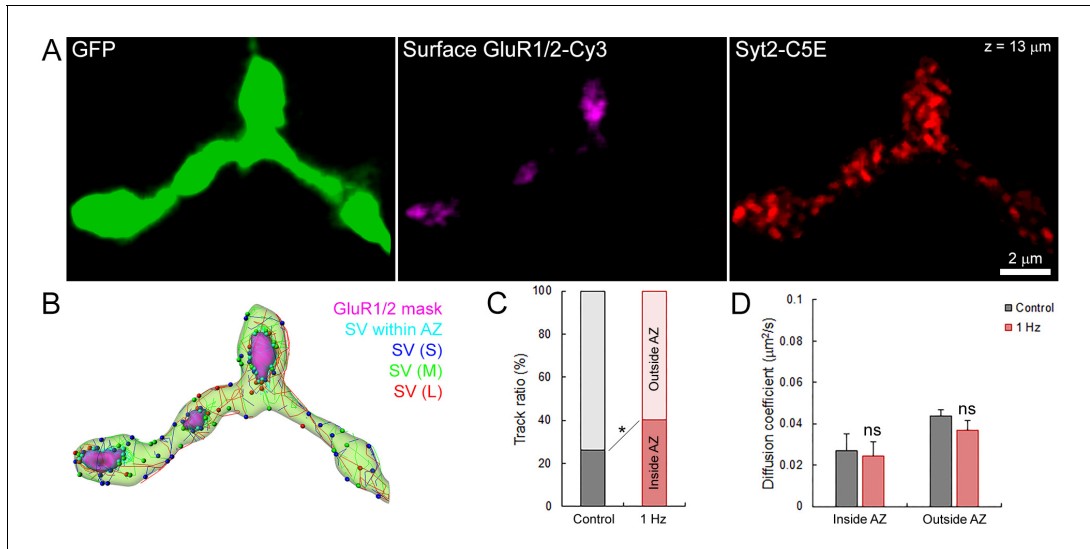

**Figure 8.** Electrical stimulation does not affect SV mobility within or outside of active zone. (**A**) Live confocal imaging of surface GluR1/2-Cy3 and Syt2-C5E-loaded SVs in GFP over-expressing giant calyceal terminal. (**B**) SV tracking color-coded according to trajectory length (Blue < 2 $\mu m$, Green 2–4 $\mu m$ and red >4 $\mu m$) and within (Cyan) AZs (Magenta). (**C**) Quantification of the number of SV trajectories inside or outside of AZs in control (Black, n = 3 terminals) and during electrical stimulation at 1 Hz for 30 s (Red, n = 3). (**D**) Comparison of diffusion coefficient of SVs inside and outside of AZs between control (Black, n = 6 terminals) and electrical stimulation at 1 Hz (Red, n = 6). Two-tailed unpaired t-test (*p<0.05; ns, not significant). Rich media files.

The following source data is available for figure 8:

**Source data 1.** Data and statistics for *Figure 8C and D*.

n = 6) and during stimulation (p=0.18, n = 6). Thus, our data indicate that electrical stimulation recruits SVs to AZs without altering their overall mobility.

## Discussion

We performed a spatio-temporal analysis of fluorescently labeled SVs in mammalian central synapses to characterize their dynamic properties and movements in presynaptic terminals. Comparative analyses of SV trajectories between (i) giant calyceal and small hippocampal terminals, (ii) morphologically mature and immature calyceal terminals and (iii) SVs over-expressing VGLUT1 or VGLUT2, together with analyses of the effects of cytoskeletal perturbation and various types of synaptic stimulation, revealed fast and heterogeneous SV movements in giant terminals, involvement of the MT cytoskeleton in inter-swelling trafficking, unchanged SV mobility in response to stimulation, and an influence of presynaptic morphology and vesicular protein composition on SV dynamics and movements.

Imaging and tracking methods, as well as various analytical approaches used in this study, permitted a detailed analysis of SV movements in nerve terminals. Automated tracking of large numbers of SVs, based on an autoregressive motion algorithm, enabled characterization of complex vesicle movements, complexity that could not be assessed by simple Brownian motion analysis. Characterization of SV dynamic properties, such as maximum speed or track length, and SV displacement modalities along individual trajectories, substantially enhanced the analysis of SV movements compared with analyses based solely on diffusion coefficients (D), where subtle changes in active vesicle dynamics may simply escape detection. In calyceal giant terminals, the diffusion coefficients calculated from MSD curves were comparable to those calculated from FRAP experiments. However, we speculate that the discrepancy between the diffusion coefficients in swelling estimated from MSD and FRAP experiments might result from the different number of interconnected fingers/swellings between terminals analyzed in each experiment that would significantly influence the proportion of long and fast SV movements. This indicates minimal bias of our automated tracking method, and allowed us to compare SV mobility in calyceal terminals with those previously reported in other types of synapses (*Holt et al., 2004*; *Rea et al., 2004*; *Lemke and Klingauf, 2005*; *Gaffield and Betz, 2007*; *Shtrahman et al., 2005*; *Rothman et al., 2016*).

Our present analysis revealed at least three groups of SVs with different dynamic properties and trajectories: intra-swelling trafficking (slow and short (S) trajectories), intermediate trafficking (moderate speed and medium (M) trajectories), and inter-swelling trafficking (fast and long (L) trajectories). We showed that SV movements are highly heterogeneous in giant calyceals, combining both diffusional and active motions. These data are in agreement with recent study showing heterogeneous SV motions in rat hippocampal neuron culture (*Joensuu et al., 2016*). Taken together Joensuu et al. and our current work demonstrated that SV movements in CNS terminals are much more heterogeneous than previously thought. Among all the individual trajectories analyzed in our study, we observed a clear, positive correlation between SV trajectory length and SV traveling speed, implying that the time required for SVs to move over short or long distances could be normalized by this mechanism. This suggests that SVs remote from active zones (AZs) could potentially reach their release sites with the same efficiency as SVs closer to AZs. These findings are complementary to a report showing that recycling SVs undergoing release at neuromuscular junctions do not have preferential spatial distributions, but are scattered randomly within nerve terminals (*Rizzoli and Betz, 2004*).

To date, numerous publications have reported a wide range of SV mobility based on analyses of diffusion coefficients obtained by different methodologies at physiological or non-physiological temperatures, and/or at different types of synapses, such as in mouse hippocampal terminals (D ~ $1\times10^{-2}$ $\mu m^2$/s at RT, D = $4$–$9\times10^{-4}$ $\mu m^2$/s; *Lemke and Klingauf, 2005*; *Shtrahman et al., 2005*), goldfish retinal bipolar cells (D = $1.5\times10^{-2}$ $\mu m^2$/s at RT; *Holt et al., 2004*), lizard retinal cone cells (D = 0.1 $\mu m^2$/s at RT; *Rea et al., 2004*), or in frog (D = $2.6\times10^{-3}$ $\mu m^2$/s) (*Gaffield et al., 2006*) or mouse (D = $5\times10^{-3}$ $\mu m^2$/s) (*Gaffield and Betz, 2007*) neuromuscular junctions. Although these data might suggest that SV mobility varies among terminal types, the lack of direct and consistent comparison between synapses raises the question of whether synapse types or morphologies really influence SV mobility. Hence, we analyzed SV movements between giant calyceal and small hippocampal terminals under the same conditions and imaging methods. We found threefold higher SV

mobility in calyceal synapses (D = $6.5 \times 10^{-2}$ μm²/s) than in hippocampal synapses (D = $2 \times 10^{-2}$ μm²/s). SV mobility tended to be higher at larger calyceal swellings, but this tendency was absent at hippocampal boutons. Thus, terminal size is clearly one of the biological factors influencing SV mobility.

Interestingly, *Peng et al. (2012)* suggested that discrepancies in SV mobility reported thus far might arise from differences in molecular composition of SVs between and/or within synapses. SV proteins are commonly divided into two classes, based on their functions: transport proteins (proton pump, VGLUT, etc.) that mediate neurotransmitter uptake, and trafficking proteins (rab, synaptotagmin, etc.) involved in SV intracellular transport and movements (*Sudhof, 1999*). VGLUT isoforms have different regional and developmental expression (*Liguz-Lecznar and Skangiel-Kramska, 2007*), and reportedly produce different release probabilities (*Weston et al., 2011*). Here, we showed that over-expression of VGLUT1 or VLGUT2 also confers different dynamic properties upon SVs and influences their mobility in calyceal presynaptic terminals. Particularly, VGLUT1-expressing SVs moved faster and traveled farther than VGLUT2-expressing SVs. Remarkably, VGLUT2-expressing vesicles lost the positive correlation between speed and distance, suggesting that their capacity to reach the AZ might be less than of VGLUT1-expressing SVs, when their initial position is distant from the AZ. Speculatively, dynamic properties of VGLUT1- and VGLUT2-expressing vesicles might result from different affinities for tethering/scaffolding proteins or molecular motors. It is likely that the ratio between SVs containing specific molecular markers, as well as their distribution and positioning within terminals, explain differences in SV mobility observed at various synapses.

We have identified a large population, ~15% of labeled vesicles, moving between presynaptic swellings at high speeds and with long, directional trajectories. This population of SVs undergoing inter-swelling exchange in giant calyceal terminals appears to be three times larger than that observed in conventional-sized hippocampal synapses (*Alabi and Tsien, 2012*) and previously designated as the 'super pool.' In hippocampal terminals, actin (*Darcy et al., 2006*) and BDNF (*Staras et al., 2010*) are reportedly involved in SV movements from the super pool. In contrast, our data in calyceal terminals revealed a significant contribution of MT networks to the transport and movements of SVs between swellings. We speculate that a large 'super pool' and presynaptic MTs may be required to coordinate fast and efficient cycling of SVs among the numerous release sites present in single giant terminals, compared to the constitutive sharing and replenishment of SVs observed in multiple individual conventional boutons.

Our data show that presynaptic morphology can significantly influence SV dynamics and movements. During development, giant terminals in culture undergo significant reorganization of their presynaptic compartments (i.e. increased surface, volume, and complexity), leading to formation of mature synaptic connections and associated with a 1.5-fold reduction in SV dynamics and movements. Although we cannot exclude the possibility that other molecular mechanisms occurring during synaptic growth and maturation regulate SV movements, the high SV mobility observed in immature terminals might arise from two independent, but integrative factors: (i) co-transport of AZ components and other synaptic proteins with SV precursors (*Okada et al., 1995*; *Yonekawa et al., 1998*; *Stagi et al., 2005*) and (ii) changes in mechanical tension during synaptic development. In immature terminals, SV mobility is high and transiently increases from stage 1 to stage 2 in parallel with increased presynaptic volume, whereas in mature terminals, SV mobility decreases significantly from stage 3 to stage 4 (see *Figure 4—figure supplement 1D*). This high and increasing SV mobility observed in immature terminals could result from the necessity to coordinate vesicle trafficking with rapid transport and delivery of synaptic components prior to establishment of stable synaptic contacts (*Bury and Sabo, 2011*). On another hand, the rise in mechanical tension associated with expansion of the presynaptic area during development (*Siechen et al., 2009*; *Ahmed et al., 2012*) could increase the probability of fast active motions, as observed in *Aplysia* neurons (*Ahmed and Saif, 2014*). After formation of stable and mature synaptic contacts during stages 3 and 4, coordinated trafficking of SVs and AZ components may diminish and mechanical tensions may lessen, simultaneously reducing active transport and SV mobility. These factors, in addition to the structural organization of the MT cytoskeleton, might also account for the decrease (1.4 times) in SV mobility observed between finger-like processes and swellings.

In giant calyceal terminals, neither chemical nor electrical stimulation increased SV mobility, in agreement with previous reports at the neuromuscular junction (*Betz and Bewick, 1992*) or at hippocampal synapses (*Lemke and Klingauf, 2005*; *Kamin et al., 2010*). These results imply that SV trafficking between endocytosis and exocytosis remain largely unchanged upon stimulation.

However, our results do not exclude the possibility that SVs, undergoing exocytosis, might transiently change their mobility during stimulation, and image analysis at higher spatial and temporal resolution might resolve putative changes in SV movements involved in neurotransmitter release. Nevertheless, our analysis has revealed some alterations of SV dynamics after KCl stimulation, inducing clustering of SVs in calyceal swellings, and a marked reduction in long trajectory SV movements. Presumably, after KCl stimulation, SVs were immobilized near release sites. Likewise, hypertonic sucrose stimulation, which depletes SVs from the RRP (*Stevens and Tsujimoto, 1995*) significantly reduced the number of actively moving SVs, suggesting that SVs depleted from the RRP during exocytosis were replenished from a recycling pool of SVs previously moving with active displacements. Direct support of synaptic transmission might be provided by fast diffusive and subtle local changes in SV mobility near release sites as recently reported (*Rothman et al., 2016*), rather than diverse and heterogenous SV movements prior to release. The latter may contribute to distribute SVs in optimal locations for the functional and structural maintenance of presynaptic terminals. In this regard, during the process of SV labeling, newly endocytosed SVs had low mobility with their distribution confined near endocytic regions for the first hour. Low SV mobility near exo/endocytic regions is likely caused by tethering of SVs around release sites. Classically, synapsin-1 is thought to tether SVs in its dephosphorylated form (*Llinás et al., 1985*). The broad-spectrum phosphatase inhibitor OA increases SV mobility by ~10 times in hippocampal terminals (*Jordan et al., 2005*) or at the neuromuscular junction (*Gaffield et al., 2006*). In calyceal terminals, OA only increased SV mobility by ~1.4 times, similar to that recently reported at cerebellar mossy fiber terminals (~2 times; *Rothman et al., 2016*), suggesting higher abundancy of untethered SVs in these terminals and/or phosphorylation independent SV tethering with molecules such as Basson (*Hallermann et al., 2010*) or Unc analogs (*Böhme et al., 2016*).

We have also demonstrated that SVs accumulate at release sites during electrical stimulation without significantly changing their mobility compared to resting condition. Although we do not exclude the possibility that SV dynamics at release sites might be affected by synaptic activity, higher temporal and spatial resolution imaging methods would be required to assess putative changes in the mobility of SVs associated with neurotransmitter release.

Altogether, our data indicate that in central nervous system, SV movements are highly heterogenous and that large synapses possess higher basal SV mobility than smaller synapses. Our results also suggest that SV movements and supply can be influenced by morphological characteristics of presynaptic terminals and by molecular signatures of vesicles. Although the underlying molecular mechanisms remain to be characterized, presynaptic morphology and vesicular composition appeared to be major biological determinants characterizing SV dynamics and trafficking in central synapses.

## Materials and methods

### Primary neuronal cultures

Giant synapse primary cultures were established as described previously (*Dimitrov et al., 2016*). Briefly, mouse brains from E18 to P1 were extracted in ice cold HBSS (Life Technologies, USA) and the cochlear nuclei (CN) and the medial nuclei of the trapezoid body (MNTB) regions were micro-dissected and stored separately in ice cold HBSS. CN and MNTB regions were dissociated using Nerve Cell Dissociation Medium (Sumitomo Bakelite, Japan) according to the manufacturer's instructions. Dissociated neurons were then plated at an equal ratio of CN and MNTB neurons to a final density of 160,000–180,000 cells per 35 mm culture dish (Ibidi, Germany), previously coated with 100 µg/ml poly-D-lysine (Millipore, USA), in 'Nerve Cell Culture Medium' (Sumitomo Bakelite, Japan) supplemented with NGF2.5S 100 ng/ml (Life Technologies, USA), hBDNF 25 ng/ml (R and D Systems, USA), hFGF2 5 ng/ml (Peprotech, USA), 50 ng/ml hNT-3 (Peprotech, USA) and 20 mM KCl (Nacalai Tesque, Japan). At DIV 8, 5 µM AraC (Sigma Aldrich, japan) was added to the medium to inhibit cell proliferation. Medium without AraC was exchanged every 4 days throughout the culture.

Hippocampal neurons were prepared as described previously (*Guillaud et al., 2008*) and dissociated hippocampal neurons were plated to a final density of 150,000 cells per 35 mm culture dish. When needed and before plating, transfection of dissociated VCN or hippocampal neurons with pCAG-AcGFP (*Dimitrov et al., 2016*), Venus-VGLUT1 or Venus-VGLUT2 vectors was performed by

electroporation using the Neon transfection system (Life Technologies, USA) according to the manufacturer's instructions.

## Antibodies

Primary antibodies: Vesicular Glutamate Transporter 1 (VGLUT1, Millipore, USA), Synaptophysin (Synaptic System, Germany), Synaptotagmin-2 lumenal domain (Synaptic System), detyrosinated α-tubulin (Glu-α-tubulin, Synaptic System), α-tubulin (Sigma Aldrich, Japan), Kinesin motor protein KIF1A (AbCam, USA), GluR1/2-Cy3 extracellular domain (BIOSS antibodies).

Secondary antibodies: AlexaFluor 405, 488, 568 and 647, DAPI, Quantum dots Q655 and Q585 (all from Life Technologies, USA). CypHer5E (C5E, Synaptic System).

## Immunofluorescence microscopy

Primary cell cultures grown in 35-mm culture dishes were fixed in PBS 4% paraformaldehyde for 20 min at room temperature or overnight at 4°C. After fixation, cells were permabilized in PBS 0.2% saponin for 12 min and blocked in PBS 3% bovine serum albumin (BSA) for 45 min at room temperature. Primary antibodies, diluted in PBS 0.02% saponin and 0.3% BSA, were incubated overnight at 4°C. Cells were then washed three times in PBS 0.02% saponin for 10 min and fluorescent secondary antibodies were incubated in PBS 0.02% saponin and 0.3% BSA for 1 hr at room temperature. After washing three times in PBS 0.02% saponin for 10 min, cells were mounted in PBS or Prolong gold antifade reagent (Life Technologies, USA). Confocal imaging was performed on a confocal laser scanning LSM780 microscope equipped with Plan-Apochromat 63x, 1.4 NA or Plan-Neofluar 100x, 1.45 NA oil immersion lenses (Carl Zeiss, Germany).

## Synaptic vesicle labeling and live imaging

Both fluorescent-conjugated antibodies directed against the luminal domain of synaptotagmin and fluorescent nanoparticles have been used to label and assess SV cycling in cultured mammalian synapses (*Kraszewski et al., 1995*; *Zhang et al., 2007*; *2009*; *Lee et al., 2012*). Here, SVs from cultured giant terminals were labeled with rabbit polyclonal antibodies directed against the intravesicular (lumenal) domain of synaptotagmin-2 tagged with either quantum dots Q655 or Q585, or with the pH-sensitive fluorophore CypHer5E (C5E). Briefly, 1 µg of synaptotagmin-2 antibody was incubated with 2.5 µg of secondary F(ab)'$_2$ antibody (Q655 or C5E) in a total volume of 10 µl for 45 min at room temperature before adding it to the culture. Synaptotagmin-2 solution was then applied to cultures between DIV 15 and DIV 21 for 1–16 hr (typically overnight) at 37°C and 5% CO$_2$. This procedure ensured that only vesicles fused to the plasma membrane during spontaneous exocytosis were labeled and we expected that the labeled vesicle pool comprised $\geq$1–2% of all vesicles found in the terminals. Before imaging, culture medium was replaced with standard Tyrode's solution (pH = 7.4) and live imaging was performed on an LSM 780 confocal laser scanning microscope equipped with a temperature-controlled (Tokai Heat, Japan) Plan-Apochromat 63x, 1.45 NA oil immersion lens (Carl Zeiss). Tyrode's solution was continuously perfused with a Dynamax peristaltic pump (Rainin, Switzerland) connected to a dual automatic temperature controller TC-334B (Warner Instruments Corp., USA), to maintain a constant physiological temperature of 36.5°C throughout the imaging period. After localizing giant presynaptic terminals over-expressing cytosolic GFP, a region of interest (ROI) containing several interconnected presynaptic swellings, located in the upper region of the calyceal terminal (z = 10–15 µm), was identified in order to achieve an effective scanning speed of 1–1.25 frame per second for up to 120 s, on single 2D optical section (initial image resolution 512 × 512 pixels, pixel dwell time 3.15 µs). For 3D tracking, four optical sections (~300 nm) covering the height of presynaptic swellings (~1.5 µm) were acquired with a pixel dwell time of 1.27 µs. Raw confocal images were filtered using the median filter algorithm in ZEN 2.1 (Carl Zeiss) before further analysis.

## Image analysis and vesicle tracking

Accurate automatic and simultaneous tracking of a large population of objects, such as vesicles, requires robust and sophisticated algorithms that can identify independent objects, predict their future positions based on their previous speed, directionality, and weighted intensity. Here, IMARIS 8.1 with IMARIS Track, Measurement Pro and Vantage plugins (Bitplane, Switzerland), commonly

used to analyze organelle movements (*Johnson et al., 2011*; *Wong and Munro, 2014*; *Grassart et al., 2014*; *Maucort et al., 2014*; *Varela et al., 2015*), was used to track SV movements and to perform spatio-temporal analyses of several thousand vesicle trajectories. Spot detection and tracking was performed on 30 s sequences using either the autoregressive or Brownian motion algorithm with an initial spot size of 150 nm (Gaussian fitted), a maximum distance between spots on two consecutive frames of 0.8 µm without frame gap. The distance between spots on two consecutive frames was set to 0.8 µm because kinesin-driven transport in mammalian neurons has an average speed of 0.7–1 µm/s (*Guillaud et al., 2003*); thus, we conservatively estimated the maximum distance travelled by a vesicle associated with kinesin within a 1 s interval would be ~0.8 µm. For rendering and visualization, synaptic vesicles were color-coded according to their respective speeds, and tracks were color-coded according to time or to their trajectory length. Analysis and comparison of SV trajectories and dynamic properties were performed in mature terminals unless stated otherwise. Statistical data sets regarding pre-synaptic volume and surface, vesicle speed and track length, etc. were exported from IMARIS Measurement Pro to Prism 6 (Graphpad software Inc., USA) and MS Excel (Microsoft, USA). The mean square displacement (MSD) analysis and estimation of the diffusion coefficient (D) from vesicle trajectories was performed using the MSD Matlab plugin for IMARIS (*Tarantino et al., 2014*). The diffusion coefficient from FRAP experiments was calculated as $D = [0.224x(\omega^2/T_{1/2})]$, where $\omega$ represents the length of the bleached region of interest and $T_{1/2}$, the time to recover half of the fluorescence intensity (*Axelrod et al., 1976*). The Pearson coefficient (P) for co-localization was calculated in IMARIS.

## Live imaging of microtubule, actin networks and AMPA receptors

Labeling of microtubule and actin networks in live terminals was performed with SiR-tubulin and SiR-actin (Cytoskeleton Inc., USA) according to the manufacturer's instructions. Briefly, primary cultures were exposed to 1 µM SiR-Tubulin or SiR-Actin in Tyrode's solution for 30 min at 37°C. Labeling solution was replaced with fresh Tyrode's solution without SiR-Tubulin or SiR-Actin before imaging.

Surface labeling of AMPA receptors in giant calyceal culture was performed by incubation with 10 µg/ml mouse monoclonal antibodies directed against the extracellular domain of GluR1/2 tagged with Cy3, for 15 min at room temperature before live imaging. Projection of the surface area labeled with GluR1/2-Cy3 on to the pre-synaptic terminal was used to delineate release sites and compare SV mobility inside and outside putative active zones.

## Drug treatments and field stimulation

De-polymerization of microtubules and disruption of actin filaments were performed by bath application of 30 µM nocodazole (Tocris, USA) or 10 µM latrunculin-A (Focus Biomolecules, USA), respectively. Okadaic acid (Tocris, USA) was used at 2.5 µM (*Betz and Henkel, 1994*; *Kraszewski et al., 1995*). Electrical field stimulation (*Ohhashi et al., 1980*) of giant terminals in culture was performed at a frequency of 1 Hz, with voltage ranging from 50 to 100 V and a pulse duration of 0.3 ms.

## Statistical analysis

No statistical method was used to predetermine sample size. Experiments were not randomized and investigators were not blinded to allocation during experiments. Each experiment was repeated three times independently to ensure reproducibility and adequate statistical power. All data sets were compared using two-tailed, unpaired Student's t-tests and two-way ANOVA in Prism 6 (Graphpad Software Inc.). Data are presented as mean ± s.e.m. pooled from at least three independent experiments. Statistical significance (*) was assumed when $p \leq 0.05$ (data sets and exact p values are provided in source data files).

## Ethical statement

All experiments have been performed in accordance to the regulations of OIST animal care and use committee (protocol #2015–128). OIST animal facilities and animal care and use program are accredited by AAALAC International (reference #1551).

## Acknowledgements

We thank Dr. Shigeo Takamori (Doshisha University, Kyoto, Japan) for the generous gift of Venus-VGLUT1 and Venus-VGLUT2 constructs and comments, Dr. Yasushi Okada (Riken QBiC, Osaka, Japan) for helpful discussions and comments, and Dr. Takeshi Sakaba (Doshisha Univeristy) for comments on our manuscript. We also thank Abdelmoneim Eshra for technical assistance in VGLUT experiments and Dr. Steven D. Aird for editing our manuscript. This work was supported by the Okinawa Institute of Science and Technology (OIST).

## Additional information

### Funding

| Funder | Author |
|---|---|
| Okinawa Institute of Science and Technology Graduate University | Tomoyuki Takahashi |

The funders had no role in study design, data collection and interpretation, or the decision to submit the work for publication.

### Author contributions

LG, Conceptualization, Data curation, Formal analysis, Investigation, Methodology, Writing—original draft, Writing—review and editing; DD, Methodology, Writing—review and editing; TT, Conceptualization, Funding acquisition, Writing—review and editing

### Author ORCIDs

Laurent Guillaud, http://orcid.org/0000-0002-9688-0991

### Ethics

Animal experimentation: All experiments have been performed in accordance to the regulations of OIST animal care and use committee (protocol #2015-128). OIST animal facilities and animal care and use program are accredited by AAALAC International (reference #1551).

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
