## [Decision Letter]

Thank you for submitting your article "Presynaptic morphology and vesicular composition determine vesicle dynamics in central synapses" for consideration by *eLife*. Your article has been reviewed by two peer reviewers, and the evaluation has been overseen by a Reviewing Editor and Gary Westbrook as the Senior Editor. The reviewers have opted to remain anonymous. The reviewers have discussed the reviews with one another and the Reviewing Editor has drafted this decision to help you prepare a revised submission.

Summary:

Guillaud et al. analyzed vesicle mobility in calyx of Held and hippocampal terminals in primary neuronal cultures using mostly quantum dot-labeled synaptotagmin-2 antibodies. The study is very detailed, and resulted in a substantial amount of descriptive observations. Movement in calyceal terminals was faster with a surprisingly high diffusion coefficient of about 0.06 µm^2^/s, and over longer stretches, than in small hippocampal synapses; this changed during calyceal maturation; vesicles expressing VGLUT1 were more mobile than vesicles expressing the similar transporter VGLUT2; removal of microtubules resulted in a loss of long-distance movement, in agreement with the current literature. Neither chemical, nor electrical stimulation ever increased the synaptic vesicle movement, again in agreement with the current literature. The work is of high technical quality (but see one concern below). It contains several interesting findings, and some results that are novel. The strongest points of the study are the direct comparison of two types of synapses within one study, the developmental analysis, and the comparison of vGlut1 and vGlut2 expressing vesicles.

Essential revisions:

1) One reviewer was concerned about the absolute size of the diffusion coefficient. This value seems to critically depend on the used tracking algorithm (0.06 µm^2^/s with autoregressive algorithm but 0.01 µm^2^/s with Brownian motion algorithm). The image acquisition rate is about 1 Hz. One worry is that the two different vesicles moving through a single optical section are erroneously tracked as one vesicle. This would also predict the longer tracks associated with faster diffusion coefficients (which is only not the case in vGlut2 expressing vesicles). The authors argue that the FRAP experiments are consistent, but (1) there are significant differences (in swellings: D = 0.022 µm^2^/s with FRAP but D = 0.041 µm^2^/s with vesicle tracking) and (2) a reliable dissection of diffusion speed and immobile fraction seems difficult (cf. increasing florescence at end of trace for swellings in Figure 3).

2) One additional experiment would substantially enhance the output of the manuscript: an experiment in which the vesicle motion is described in relation to active zones. Since the pre- and postsynaptic active zones tend to be aligned, it is irrelevant whether the former or the latter are labeled. The authors should try to label postsynaptic glutamate receptors using antibodies directed against their extracellular domains, which are commercially available, and perform some vesicle imaging experiments, with or without stimulation. This type of experiment would add quite important information, relating to the behavior of vesicles near active zones, or away from them. For example, it is conceivable that the movement near active zones is quite different during stimulation, in comparison to the resting state. Such information would make this manuscript even more interesting and useful.

---

## [Author Response]

Essential revisions:

1) One reviewer was concerned about the absolute size of the diffusion coefficient. This value seems to critically depend on the used tracking algorithm (0.06 µm^2^/s with autoregressive algorithm but 0.01 µm^2^/s with Brownian motion algorithm). The image acquisition rate is about 1 Hz. One worry is that the two different vesicles moving through a single optical section are erroneously tracked as one vesicle. This would also predict the longer tracks associated with faster diffusion coefficients (which is only not the case in vGlut2 expressing vesicles).

To address this question, we compared SV trajectories in the same terminals from times series acquired at 0.5 s per image (2x faster), 1 s per image (standard) and 2 s per image (2x slower). We have limited our data acquisition to a maximum of 0.5 s per image in order to maintain the size of the ROI and to maintain the quality of the spot detection. Acquisition faster than 0.5 s per image substantially reduced the spatial resolution and detection of individual spots unless reducing the ROI significantly, which would in turn compromise the detection of inter-swelling movements. These new results are presented in Figure 2—figure supplement 3, and reported in the third paragraph of the subsection “Fast and heterogenous vesicles mobility in giant calyceal terminals”. We showed that the proportions of slow/short (S), medium (M) and fast/long (L) trajectories were not significantly affected by the data acquisition rate, indicating that the long and fast directional runs are not likely resulting from the erroneous detection of different SVs moving through the focal plan during image intervals.

We also previously thought about the possibility that 2 different SVs moving in and out of the focal plan could also be tracked as one, and affected the detection and accuracy of SV trajectories. We thus compared SV movements from 2D (xy) and 3D (xyz) data and showed that SV trajectories and mobility were not affected either. These results are now presented in Figure 2—figure supplement 2 and more explicitly described in the aforementioned paragraph.

Although we cannot exclude the possibility of erroneous detection in our automatic tracking algorithm, the population of fast/long directional movements is highly specific as these trajectories were not only absent from VGLUT2 expressing SVs, but also significantly reduced in mature calyceal terminals, in calyceal terminal treated with nocodazole as well as in hippocampal terminals. These long and fast trajectories were also absent from the movements of SVs newly endocytosed either spontaneously or during electrical stimulation. Thus, if the proportion of these fast/long trajectories was solely due to the erroneous detection of 2 different SVs as one, one would argue that the probability of such events would occur in all conditions and thus would fail to detect the significant differences mentioned above.

In addition we would like to mention that the tracking algorithm used in IMARIS, discriminate SV based on a quality index estimated from Gaussian fitting of the fluorescence spot. If the quality indexes of 2 neighboring spots are significantly different, these 2 spots will not be traced as one.

The authors argue that the FRAP experiments are consistent, but (1) there are significant differences (in swellings: D = 0.022 µm^2^/s with FRAP but D = 0.041 µm^2^/s with vesicle tracking) and (2) a reliable dissection of diffusion speed and immobile fraction seems difficult (cf. increasing florescence at end of trace for swellings in Figure 3).

A) We have also noticed and acknowledged this discrepancy between the diffusion coefficient of SVs in swellings estimated from MSD curves and calculated from FRAP experiments in the fifth paragraph of the subsection “Mobility of synaptic vesicles in giant calyceal terminals”. We speculate that the morphological features of the terminals used in FRAP experiments and tracking experiments might have been different. A difference in the number of interconnected fingers/swelling could potentially affect the proportion of fast/long trajectories in the ROI in FRAP or tracking experiments, and might result in the observed differences in the estimation of D. Such morphological differences could rarely be found in fingers-like areas, thus reducing the possible discrepancy between FRAP and tracking experiments. We have included this discussion in the second paragraph of the Discussion.

B) Figure 3 was truncated to the first 30 s of the recovery experiments preventing the clear visualization of the full fluorescence recovery in the swelling areas. However, we calculated T_1/2_ of the fluorescence recovery and estimated the mobile/immobile fractions in swelling after the fluorescence recovered to its maximum (plateau). Thus, we replaced the truncated figure with the full time-scale fluorescence profile on new Figure 1—figure supplement 2.

2) One additional experiment would substantially enhance the output of the manuscript: an experiment in which the vesicle motion is described in relation to active zones. Since the pre- and postsynaptic active zones tend to be aligned, it is irrelevant whether the former or the latter are labeled. The authors should try to label postsynaptic glutamate receptors using antibodies directed against their extracellular domains, which are commercially available, and perform some vesicle imaging experiments, with or without stimulation. This type of experiment would add quite important information, relating to the behavior of vesicles near active zones, or away from them. For example, it is conceivable that the movement near active zones is quite different during stimulation, in comparison to the resting state. Such information would make this manuscript even more interesting and useful.

As suggested, we labeled surface AMPA receptors with fluorescently-labeled antibody against GluR1/2 extracellular domain, and compared SV mobility within or outside of AZ between resting conditions and electrical stimulation at 1Hz. We positioned putative AZs by projecting a mask corresponding to the surface area of GluR1/2-Cy3 onto the presynaptic terminal. Number and mobility of SVs within and outside of AZs were analyzed in control and during electrical stimulation. Our data showed that electrical stimulation significantly increased the number of SVs detected near AZ compared to resting condition (p = 0.018, n = 3 terminals). The mobility of SVs at AZ appeared to be reduced compared to SVs away from AZ, however this decrease was not statistically significant between control (p = 0.078, n = 6 terminals) and electrical stimulation (p = 0.179, n = 6). Similarly, SV mobility at AZ in resting condition (D = 0.027 µm^2^/s) or under electrical stimulation (D = 0.024 µm^2^/s) remained largely unchanged (p = 0.803, n = 6). These results are presented in new Figure 8, in the last paragraph of the subsection “Effects of various stimulations on SV movements in giant calyceal terminals” and in the ninth paragraph of the Discussion.

Although we did not observe significant change in SV dynamics at AZ in our current imaging setup, we do not exclude the possibility that the mobility of SVs undergoing release at AZ may be affect by synaptic stimulation. Imaging at higher temporal and spatial resolution would likely be required to analyze SV motions at AZ more accurately.